# DYANA: BENCHMARKING DYNAMIC HAND INTELLIGENCE

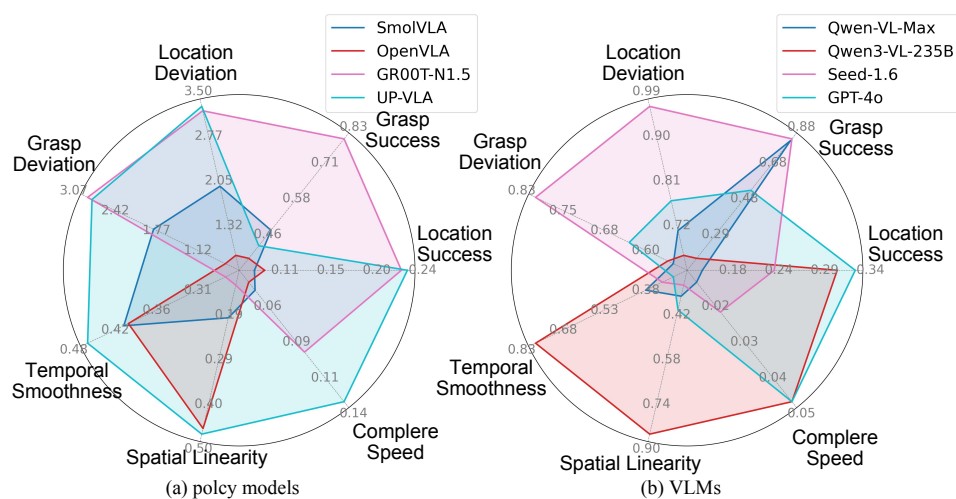

(a) polcy models  (b) VLMs

Figure 1: **The performance of SOTA models.** (a) policy models and (b) VLM models on our benchmark across seven evaluation metrics. Higher values indicate better performance after normalization (error-type metrics have been reversed for comparability).

## ABSTRACT

Most existing hand grasping benchmarks focus on static objects, which fails to capture the challenges of dynamic, real-world scenarios where targets move and precise timing becomes critical. We first propose the **Dynamic Grasp Suite (DGS)**, a unified platform for dynamic grasp evaluation, and **Dyana-12M**, a large-scale benchmark with **12M** frames of human-hand dynamic grasp trajectories. Dyana-12M represents target motion with three interpretable trajectories: straight-line, circular-arc, and simple-harmonic, which compose into arbitrarily complex trajectories. DGS standardizes interfaces and protocols, supporting the evaluation of three major model zoo: vision–language–action (VLA) agents, diffusion policies, and vision–language models (VLMs). Together, DGS and Dyana-12M establish a new paradigm for dynamic grasping, shifting evaluation from static scenes to motion-aware, temporally aligned assessment at scale.

## 1 INTRODUCTION

Hand–object interaction in the real world is dynamic: targets translate, rotate, and undergo periodic or transient perturbations. Although existing hand action prediction models have demonstrated strong performance on various benchmarks, nearly all benchmarks focusing on action prediction (Mees et al., 2022; Walke et al., 2023; Liu et al., 2023; Li et al., 2022) are limited to interactions with static objects. They are no longer match practical needs or the growing capability of modern foundations (VLMs, VLAs, diffusion policies). This leaves a blind spot: current benchmarks say little about whether state-of-the-art models can anticipate motion, act under latency, or recover from disturbances. We aim to close this gap by benchmarking dynamic hand intelligence.

To make dynamic grasping measurable and fair, we develop the **Dynamic Grasp Suite (DGS)**, a unified, online, closed-loop evaluation platform for dynamic hand–object interaction (see Figure 2). DGS exposes parameterized target-motion generators and a standardized RGB observation–action API, supports up to 7 DoF motion control for the target and two hand embodiments, and offers both 3-dim location and 15-dim rotation control channels so that both policy agents (VLA, diffusion policies) and general VLMs can be evaluated under the same protocol. Episodes run at a fixed physics step; success, endpoint errors, quality scores, and timing/robustness are computed via event-triggered criteria, no framewise hard alignment or MoCap post-processing is required.

On top of DGS we construct **Dyana-12M** (see Figure 3), a large-scale benchmark with **12M frames** across **180K** dynamic grasp trajectories dedicated to moving-target grasping. Guided by signal-theoretic and geometric views, we define three interpretable *motion primitives* **straight-line**, **circular-arc**, and **simple-harmonic**, that compose into arbitrarily complex trajectories via a compact motion-token vocabulary. Dyana-12M supports two rollout formats: (1) action-only control (3-dim /15-dim) and (2) observation + action sequences to assess observe-before-act behavior. We provide standardized evaluation for three model families: **vision–language–action (VLA)** agents, **diffusion policies**, and **vision–language models (VLMs)** used for control, with controllable difficulty and consistent reporting. In brief, our main contributions are summarized as follows:

1. **DGS platform.** A unified, hardware-agnostic, online evaluation suite for dynamic grasping with parameterized motion generators, standardized APIs, and event-triggered metrics for success, spatial accuracy, temporal alignment (lookahead/offset), trajectory quality, and robustness.
2. **Dyana-12M benchmark.** A large-scale dynamic grasp benchmark (12M frames / 180K trajectories) spanning linear/arc/harmonic motions, supporting both action-only and observation+action rollouts.
3. **Extensive model zoo.** It enables apples-to-apples comparison across VLA, diffusion, and VLM-based controllers.

## 2 RELATED WORK

### 2.1 BENCHMARKS FOR EMBODIED GRASPING

Large-scale grasping datasets such as GraspNet-1Billion enable robust learning and standardized evaluation for *static* tabletop grasping, but they assume fixed-base arms interacting with stationary objects and provide no protocol for online dynamics (Fang et al., 2020). Target-driven benchmarks have begun to stress occlusion and clutter (*e.g.*, TARGO) while still focusing on static or quasi-static targets and single-episode image–action scoring without temporal alignment (Xia et al., 2024). Efforts explicitly aimed at dynamic grasping include *DG-Bench*, which offers a reproducible setup with moving objects and visual servoing policies, but remains constrained to eye-in-hand sensing on tabletop arms and simple motion patterns (Burgess-Limerick et al., 2022). General-purpose embodied benchmarks ManiSkill (Mu et al., 2021) and BEHAVIOR-1K (Li et al., 2022) expand scene and task diversity and promote mobile manipulation, yet they neither specify parameterized target-motion generators nor provide closed-loop evaluation protocols or metrics tailored to the perception–prediction–control alignment required for grasping moving targets. Emerging whole-body settings (*e.g.*, DQ-Bench for legged manipulators) introduce non-planar terrain and 6-DoF target motion, but target a different embodiment and do not provide hand-centric dynamic grasp suites for general policies. As for the human hand grasping, **DGS** contributes a motion-controlled evaluation suite with parameterized generators (linear/circular/harmonic), a unified observation–action API that is hardware-agnostic, and metrics that capture temporal alignment (lookahead/completion-frame offset), trajectory quality, success, and robustness, enabling closed-loop dynamic grasp evaluation at scale. **Dyana-12M** further supplies 12M human-hand dynamic trajectories with controllable difficulty for reproducible training and stress testing.

### 2.2 DYNAMIC GRASPING: PLANNING AND LEARNING

Classical trajectory-optimization planners (CHOMP, ITOMP, TrajOpt) generate smooth, collision-free motions and support replanning (Ratliff et al., 2009; Park et al., 2012; Schulman et al., 2013), yet they face latency and warm-start limitations under rapid target motion and sensor occlusion, making

end-to-end timing alignment difficult (Zucker et al., 2013; Park et al., 2012). Learning-based dynamic grasping addresses reactivity by coupling prediction and control. Reachability-/motion-aware grasp selection filters candidates in real time as targets move (Akinola et al., 2021). RL-based methods like GAP-RL (Xie et al., 2025) incorporate fast grasp encoders and region exploration to handle continuous motion and sim-to-real deployment, but typically evaluate in simulator-defined scenes without a standardized, *open* dynamic-grasp protocol. For high-speed exchanges, dexterous throw-and-catch and bimanual handovers demonstrate reactive, closed-loop control, yet focus on specific interaction scripts rather than a general dynamic-grasp suite with parameterized motion generators and multi-metric reporting (Huang et al., 2023; Wang et al., 2025). Unlike prior work that (1) assumes static/quasi-static targets, (2) limits dynamic motion to tabletop or single-sensor regimes, or (3) studies specialized exchanges, Dyana unify *dynamic-target modeling* → *online, closed-loop interaction* → *standardized metrics* into a reproducible pipeline. Dyana yields interpretable control of difficulty and principled composition of complex trajectories, while the evaluation protocol removes framewise rigid alignment, directly measuring temporal anticipation and control lag.

## 3 BENCHMARK

### 3.1 PRELIMINARIES

We study Human–Object Interaction (HOI) with *moving* objects and seek a motion representation that is compact, compositional, and tunable in difficulty. We encode targets as sequences of three atoms, L (straight), A (arc), H (harmonic), which DGS uses to synthesize scenes and drive metrics. To justify this choice, we introduce minimal trajectory notation (position/velocity/acceleration and curvature) and invoke two complementary views: a signal view (Fourier components for periodicity/drift) and a geometric view (piecewise constant curvature yields lines/arcs, with clothoids for smooth transitions).

**Notation.** A trajectory $\mathbf{r} : [0, T] \to \mathbb{R}^d$ has velocity $\mathbf{v} = \dot{\mathbf{r}}$ and acceleration $\mathbf{a} = \ddot{\mathbf{r}}$. For planar motion,

$$\kappa(t) = \frac{\|\mathbf{v}(t) \times \mathbf{a}(t)\|}{\|\mathbf{v}(t)\|^3},$$

and with arc length $s$, the curve's shape is determined (up to a rigid motion) by $\kappa(s)$.

**Signal-theoretic view (concise).** Each coordinate of $\mathbf{r}(t)$ is a time signal, and periodic components admit a finite Fourier approximation $x(t) \approx a_0 + \sum_{k=1}^{K} (a_k \cos k\omega t + b_k \sin k\omega t)$. A sine–cosine pair yields circular motion; the affine term yields linear drift. This motivates a compact H (harmonic) token defined by dominant frequency/phase over local windows.

**Geometric view (concise).** Approximating curvature $\kappa(s)$ by steps gives piecewise constant-curvature paths, straight segments and circular arcs, consistent with Dubins/Reeds–Shepp shortest-path structures. Clothoids provide $G^2$ connectors when needed (Dubins, 1957; Reeds & Shepp, 1990; LaValle, 2006). This justifies L/A (and optional C) tokens with interpretable length/radius/curvature-slope parameters. More details please refer to Section A.6

### 3.2 SCENE AND TASK DESIGN

This work focuses on the task of **Dynamic Localization & Grasping**, as illustrated in Figure 2. The target object moves continuously within the $XZ$ plane and follows three representative atomic trajectories: linear, circular, and harmonic oscillation. Different instances are generated by parameterization such as velocity, radius, amplitude, and frequency. The agent is a five-fingered hand with 18 degrees of freedom. It must infer the motion pattern of the object after observing sufficient visual evidence and then complete localization and grasping of the moving target.

The scene provides egocentric RGB images $I_t \in \mathbb{R}^{H \times W \times 3}$ from a fixed camera. All spatial quantities are defined in the camera coordinate system ($Z$ forward, $X$ right, $Y$ up). At each time step $t$, the agent receives the observation

$$o_t = \{I_t, h_t, q_t, k_t, x^{\text{text}}\},$$

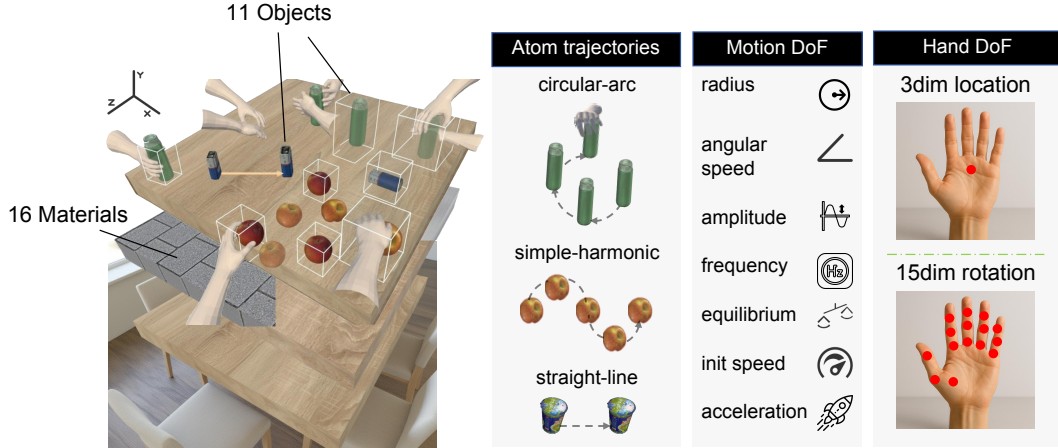

Figure 2: **The overview of Dynamic Grasp Suite (DGS).** A unified dynamic grasp platform composed of three atom trajectories with 7 motion DoF and 2 hand DoF of 3 3-dim end-effector location channel and a 15-dim rotation/pose channel.

where $h_t$ is the 6D pose of the palm center, $q_t$ the poses of 15 joints, $k_t$ the poses of six keypoints (five fingertips and the wrist), and $x^{\text{text}}$ the task description.

The policy $\pi$ predicts an 18-dimensional control vector from the history of observations $o_{0:t}$:

$$a_t = (a_t^{\text{loc}}, a_t^{\text{gra}}),$$

where $a_t^{\text{loc}} \in \mathbb{R}^3$ denotes the localization action of the palm center, and $a_t^{\text{gra}} \in \mathbb{R}^{15}$ denotes the joint rotation angles of the 15 fingers.

### 3.3 DYNAMIC GRASP SUIT

**Dynamic grasp trajectory generation.** DGS provides (1) automated motion and grasping components that generate trajectories for arbitrary objects, motion types, and parameters; (2) automated hand grasping logic for dynamic objects that ensures accurate localization and grasping; and (3) an automated data collection pipeline that enables large-scale trajectory generation. See Section A.4 for details.

**Unified Evaluation Interface.** As shown in Figure 3, DGS incorporates three categories of mainstream models into a single evaluation framework, including **VLA models**, **diffusion policy models**, and **VLMs**. The interface standardizes the observation–action process into a unified data flow: the input consists of RGB visual information and task description, and the output is hand localization in 3D space together with 15 joint angle controls. For models from different domains (e.g., VLA models designed for robotic arms or diffusion policy models for dexterous hands), DGS performs lightweight finetuning to align their output space with the 18-dimensional hand control parameters.

**Observe-before-act Mechanism.** Unlike static grasping, dynamic grasping emphasizes temporal prediction and anticipation. To this end, DGS introduces "Observe-before-act" format in both data collection and evaluation protocols. As show in the middle panel of Figure 3, the hand remains fixed while the object continues to move according to predefined dynamics. This window allows the model to perceive temporal variations and learn motion patterns. In contrast, the traditional direct-act approach proceeds directly to action execution, lacking explicit temporal modeling and often leading to delayed responses or prediction errors. DGS supports both **observe-before-act** and **direct-act** formats for data collection and online evaluation, while accommodating evaluation for both static and dynamic grasping tasks.

**Hierarchical Evaluation Metrics.** In dynamic localization and grasping, single-dimensional evaluation cannot capture overall model performance. We design a hierarchical metric system with three

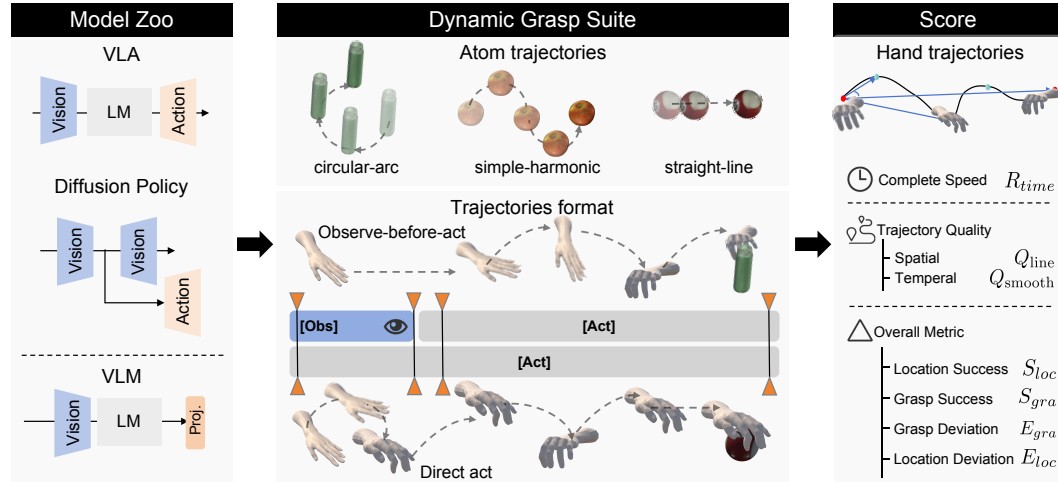

Figure 3: **The framework of Dyana benchmark.** Model zoo with three families, **VLA**, **diffusion policies**, and **VLM**-based controllers. *Middle:* Dynamic Grasp Suite provides atom trajectories (circular-arc, simple-harmonic, straight-line) and two rollout formats: *observe-before-act* and *direct act*. *Right:* Multi-dimensional scoring includes runtime $R_{time}$, trajectory quality (spatial $Q_{line}$, temporal $Q_{smooth}$), and overall metrics (location/grasp success $S_{loc}$, $S_{grasp}$; deviations $E_{loc}$, $E_{grasp}$).

levels: overall success, trajectory quality, and completion speed. Details of metric design and formula definitions are provided in Section 4.4.

### 3.4 DYANA-12M BENCHMARK

**Data collection.**  The Dyana-12M benchmark dataset is constructed on top of the DGS platform. To enhance data diversity, we design multiple interception-time settings (0.5s, 1.0s, and 1.5s), corresponding to fast, medium, and slow grasping rhythms. This allows objects under the same motion state to be grasped in different ways, encouraging models to learn to execute grasping actions at multiple time points within the motion window.

During data collection, trajectories are recorded at 20 FPS. Each frame contains an egocentric RGB image captured by a camera oriented toward the motion plane, along with synchronized human-hand state information generated by the Unity backend. This includes the 6D pose of the palm center $h_t$, the 6D poses of 15 controllable joints $q_t$, and the 6D poses of five fingertip keypoints together with the wrist joint $k_t$, with all rotations represented in Euler angles.

To ensure trajectory quality, we strictly constrain the arrival time difference between the hand and the object at the interception point to within 0.05s (*i.e.*, one frame). Moreover, beyond the observation stage, the global hand position and joint angles are enforced to evolve smoothly and continuously over time, without introducing stagnation points.

**Datasets statistics.**  Dyana-12M is characterized by both large scale and high density, containing nearly 200K episodes and about 12M image frames, spanning a total duration of approximately 600K seconds (over 160 hours). The dataset also exhibits substantial diversity in objects and task types: it includes 11 object categories covering spherical, cylindrical, and box-like geometries, among which 6 objects possess self-rolling properties, further increasing interaction complexity. Task settings encompass three fundamental motion types (linear, circular, and harmonic), with tens of thousands of parameter configurations generated for each, ensuring both richness and coverage of trajectory patterns.

As illustrated in Figure 4(a), Dyana-12M incorporates a clear hierarchy of dynamic task difficulty, ranging from simpler periodic motions (circular and harmonic oscillations) to more complex non-periodic motions. Specifically, the dataset contains over 184K trajectory instances, with circular

| Benchmark | Protocol | | Embodiment | | Scale | |
|---|---|---|---|---|---|---|
| | Moving obj. | Obs. phase | Free-floating | Agent DoF | Frames | Trajectories |
| Meta-World (McLean et al., 2025) | ✗ | ✗ | ✗ | 7, 9 | – | – |
| BridgeData V2 (Walke et al., 2023) | ✗ | ✗ | ✗ | 7 | – | 60K |
| LIBERO (Liu et al., 2023) | ✗ | ✗ | ✗ | 7 | 1.0M | 6.5K |
| CALVIN (Mees et al., 2022) | ✗ | ✗ | ✗ | 7 | 2.4M | – |
| RoboMIND (Wu et al., 2025) | ✗ | ✗ | ✗ | – | – | 107K |
| VinT-6D (Wan et al., 2024) | ✗ | ✗ | ✗ | 8, 16 | 2.1M | – |
| DexArt (Bao et al., 2023) | ✗ | ✗ | ✗ | 16 | – | – |
| **Ours (DGS/Dyana)** | ✓ | ✓ | ✓ | **18** | **12.0M** | **180K** |

Table 1: **Scope comparison with popular manipulation datasets/benchmarks.** Most prior resources lack *moving* targets, a dedicated *observation* phase for online evaluation, or a *free-floating* (hand-centric) embodiment. **DGS/Dyana** provides all three and scales to **12M** frames and **180K** trajectories. Dashes indicate values not reported by the original papers.

and harmonic motions accounting for 48.6% and 27.3% respectively, while the more challenging linear motions account for 24.1%. The diversity of the benchmark primarily stems from parameter variations: linear trajectories differ in direction, velocity, and acceleration, while circular trajectories vary in radius and angular velocity. As shown in Figure 4(b), linear, harmonic, and circular motions correspond to 23,520, 27,021, and 44,508 parameter sets respectively, yielding nearly 100K distinct trajectory modes. Combined with 11 object categories, this results in a wide coverage of complex hand–object interaction combinations.

In terms of episode length, as shown in Figure 4(c), circular and harmonic motions have average lengths of 78 and 64 frames respectively, constituting the majority of the dataset. In contrast, linear motions average only 36 frames. Due to the absence of periodic regularities, these tasks pose greater challenges for models, requiring them to infer motion trends rapidly within short observation windows and to make timely grasping decisions.

**Compare to other benchmarks.**

As shown in Table 1, mainstream hand–object interaction datasets provide fine-grained pose capture and generation, with high-precision annotations of hands and joints. However, they are limited to ensuring the physical plausibility of hand-object interactions and lack explicit perception of object positions in 3D space. As a result, they cannot evaluate dynamic perception or prediction. Moreover, these benchmarks require extensive manual annotation and cannot be expanded once collected. Their evaluation is usually offline, focusing on reconstruction and imitation rather than policy generation or temporal decision-making. Since target objects remain static, task complexity mainly comes from hand pose fitting rather than reasoning in motion environments. Thus, HOI datasets are better suited for low-level perception and reconstruction, but inadequate for assessing generalization in dynamic grasping scenarios.

VLA benchmarks span multi-task, multimodal manipulation but largely assume *static* objects and fixed-base arms with hinge-joint commands, an indirect proxy for spatial reasoning. They also lack an explicit *observation* phase: agents act from the initial state without first inferring target dynamics, which suffices for static scenes but bypasses real-time perception–prediction–control. Consequently, like HOI datasets, they do not probe spatio-temporal reasoning under motion. **Dyana-12M** closes this gap by introducing moving targets and an explicit observe→act protocol with dexterous, high-DoF hand control and large-scale, diverse trajectories for dynamic grasping.

## 4 EXPERIMENTS

### 4.1 EXPERIMENT SETUP

All experiments are conducted on the proposed **Dynamic Grasp Suite (DGS)**, which is implemented on the Unity engine (Editor version 2022.3.58f1c1). For reproducibility, the internal clock is disabled and the simulator advances with fixed-step intervals to ensure consistent timing. Model training and evaluation are deployed on a distributed GPU cluster with CUDA 12.4 and $4 \times$ A100

| Benchmark | Protocol | | Data | Scope | Labels |
|---|---|---|---|---|---|
| | Online eval. | Moving objects | Automated collect. | Obj./task variability | Human annotation |
| ARCTIC (Fan et al., 2023) | ✗ | ✗ | ✗ | ✗ | ✓ |
| GRAB (Taheri et al., 2020) | ✗ | ✗ | ✗ | ✗ | ✓ |
| GigaHand (Fu et al., 2025) | ✗ | ✗ | ✗ | ✗ | ✓ |
| **Ours (DGS/Dyana)** | ✓ | ✓ | ✓ | ✓ | ✗ |

Table 2: **Comparison of hand–object benchmarks on dimensions critical for *dynamic* grasping.** Prior datasets emphasize static scenes and human-labeled poses, lacking *online* closed-loop evaluation with *moving* targets and automated data pipelines. **DGS/Dyana** provides parameterized motion generators, a unified observation–action API, and *event-triggered* success/trajectory metrics that avoid framewise alignment and remove manual annotation.

GPUs (80GB). All models are reproduced following the official open-source repositories, including training pipelines, architectures, and hyperparameters. To ensure fair comparison, only action-related components are minimally fine-tuned to adapt to our hand control outputs.

## 4.2 EVALUATION METRICS

**Overall metrics.** This level directly reflects whether the task is completed as well as the overall error.

- **Localization success rate** ($S_{loc}$): the ratio of trajectories where the hand successfully reaches above the target object.
- **Grasping success rate** ($S_{gra}$): the ratio of trajectories where the hand executes a valid grasp.
- **Localization error** ($E_{loc}$): the average shortest distance in the $XZ$ plane between the palm center and the target object during execution.
- **Grasping error** ($E_{gra}$): the average shortest distance between fingertips and the target surface at completion.

**Trajectory quality metrics.** For successful trajectories, we evaluate temporal smoothness and spatial linearity.

For **temporal smoothness**, given a trajectory with $N$ points $p_t \in \mathbb{R}^3$, displacement between consecutive points is $d_t = \|p_{t+1} - p_t\|_2$. We measure step consistency using the coefficient of variation ($\mathcal{CV}$):

$$Q_{\text{smooth}} = \frac{1}{1 + \mathcal{CV}((d)}, \quad \mathcal{CV}((d)) = \frac{\sigma_d}{\mu_d}, \quad Q_{\text{smooth}} \in (0, 1].$$

Here, $\mu_d$ and $\sigma_d$ denote the mean and standard deviation of $\{d_t\}$. Smaller variation yields higher $Q_{\text{smooth}}$, indicating smoother trajectories.

For **spatial linearity**, we consider the overall trajectory direction $\hat{v}$ as the normalized vector from the starting point $p_1$ to the end point $p_N$, while each segment direction $\hat{s}_t$ is defined as the normalized displacement from $p_t$ to $p_{t+1}$. The trajectory linearity is then computed as

$$Q_{\text{line}} = \frac{1}{N-1} \sum_{t=1}^{N-1} \hat{s}_t \cdot \hat{v}, \quad Q_{\text{line}} \in [-1, 1].$$

A value close to 1 indicates a highly efficient and straight trajectory, whereas lower values suggest unnecessary curvature or deviation from the overall direction.

**Completion speed metric.** We also evaluate task efficiency. Let $N$ be the total frames in the test video and $T$ the frame index at task completion. The time score (Higher scores indicate faster completion) is:

$$R_{\text{time}} = 1 - \frac{T}{N}, \quad R_{\text{time}} \in [0, 1].$$

| Model | Architecture | Params | Localization | | Grasping | | Trajectory Quality | | Runtime |
|---|---|---|---|---|---|---|---|---|---|
| | | | $S_{loc}$ (%) ↑ | $E_{loc}$ ↓ | $S_{gra}$ (%) ↑ | $E_{gra}$ ↓ | $Q_{smooth}$ ↑ | $Q_{line}$ ↑ | $R_{time}$ ↑ |
| GT | — | — | 100.00 | 0.16 | 100.00 | 0.09 | 0.81 | 0.96 | 0.16 |
| OpenVLA | AutoRegressive | 7B | 9.25 | 3.26 | 37.20 | 2.85 | **0.54** | **0.71** | 0.04 |
| UP-VLA | | 1.6B | **22.60** | **0.84** | 41.60 | 0.75 | 0.47 | 0.38 | **0.13** |
| GR00T | Diffusion | 2.7B | 22.04 | 0.91 | **79.00** | **0.68** | 0.28 | 0.26 | 0.09 |
| SmolVLA | | 500M | 8.30 | 2.14 | 47.00 | 1.72 | **0.54** | 0.11 | 0.05 |

Table 3: **Online evaluation of mainstream policy models on DGS.** $S_{loc}/S_{gra}$ are success rates (higher is better), $E_{loc}/E_{gra}$ are mean endpoint errors (lower is better). $Q_{smooth}$ and $Q_{line}$ measure trajectory smoothness and line-adherence (higher is better). $R_{time}$ is normalized runtime per episode (lower is better). **Bold** / underline denote best / second-best among learned policies; the *GT* row is an oracle upper bound.

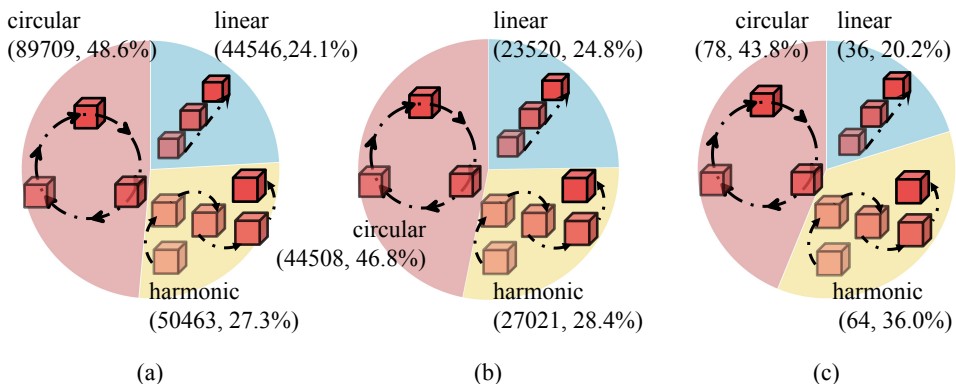

Figure 4: Statistics of the dynamic grasp dataset. (a) The number of episodes corresponding to each motion type. (b) The number of unique parameter configurations for each motion type. (c) The average video length for each motion type.

### 4.3 EVALUATION ON POLICY MODEL

We evaluate four representative policy models capable of directly outputting action sequences on the DyanaHand-12M Benchmark test-1K set. The models include two autoregressive models, Open-VLA (Kim et al., 2024) and UP-VLA (Zhang et al., 2025), and two diffusion-based models, GR00T-N1.5 (NVIDIA et al., 2025) and SmolVLA (Shukor et al., 2025).

From Table 3, all models perform significantly below the GT level in overall success rate. OpenVLA and SmolVLA achieve localization success rates below 10%, failing to reliably identify moving objects in space. For SmolVLA, this likely stems from its limited parameter scale, which constrains its ability to learn complex grasping strategies. OpenVLA's weakness arises from its one-frame-one-prediction paradigm, which lacks global trajectory modeling. Even UP-VLA, the best-performing model, achieves only 22.60% localization success, with grasp success reate still above 41.6%.

In terms of trajectory quality, all models show large gaps compared to GT. Even the best results yield trajectory smoothness $Q_{smooth}$ below 0.6, and spatial linearity $Q_{line}$ remains in the 0.1–0.4 range, except for OpenVLA. Besides, GR00T-N1.5 achieves the lowest score in $Q_{smooth}$ and the second-lowest in $Q_{line}$, while SmolVLA performs worst in linearity. These results indicate that both models produce actions with noticeable oscillations. A possible reason is that diffusion-based models, although capable of predicting multiple frames in one window, fail to adequately model trajectory consistency across windows. In contrast, autoregressive models such as OpenVLA and UP-VLA better preserve both spatial linearity and temporal smoothness.

| Model | Architecture | Params | Localization | | Grasping | | Trajectory Quality | | Runtime |
| | | | $S_{loc}$ (%) ↑ | $E_{loc}$ ↓ | $S_{gra}$ (%) ↑ | $E_{gra}$ ↓ | $Q_{smooth}$ ↑ | $Q_{line}$ ↑ | $R_{time}$ ↑ |
|---|---|---|---|---|---|---|---|---|---|
| GT | — | — | 100.00 | 0.16 | 100.00 | 0.09 | 0.81 | 0.96 | 0.16 |
| Qwen-VL-Max | | — | 15.00 | 0.91 | 80.00 | 0.80 | 0.38 | 0.35 | 0.01 |
| Qwen3-VL | AutoRegressive | 235B | 30.00 | 0.96 | 16.00 | 0.79 | **0.78** | **0.85** | **0.05** |
| Seed-1.6 | | — | 23.00 | **0.66** | **81.00** | **0.55** | 0.32 | 0.31 | 0.02 |
| GPT-4o | | — | **32.00** | 0.85 | 53.00 | 0.72 | 0.28 | 0.40 | **0.05** |

Table 4: **Online evaluation of mainstrea VLMs on DGS.** $S_{loc}/S_{gra}$ are success rates (higher is better), $E_{loc}/E_{gra}$ are mean endpoint errors (lower is better). $Q_{smooth}$ and $Q_{line}$ measure trajectory smoothness and line-adherence (higher is better). $R_{time}$ is normalized runtime per episode (lower is better). **Bold** / underline denote best / second-best among learned policies; the *GT* row is an oracle upper bound.

Overall, current policy models remain unsatisfactory for dynamic localization and grasping. The core issue is their reliance on single-frame images and states to predict future actions, while neglecting temporal motion patterns in historical frames. In dynamic environments, object trajectories exhibit strong temporal dependencies. Without effective modeling of such patterns, stable and accurate grasping is difficult.

### 4.4 EVALUATION ON VLM MODEL

We evaluate four mainstream large VLMs, Qwen-VL-Max, Qwen3-VL, Seed-1.6, and GPT-4o. Except for Qwen3-VL, all models are closed-source. They are tested on the same benchmark set.

As shown in Table 4, mainstream VLMs still show significant gaps from GT in dynamic grasping tasks. For localization success rate ($S_{loc}$), GPT-4o (32%) and Qwen3-VL (30%) perform relatively better, but remain far below static-scene levels. This indicates that current VLMs have limited ability to model and predict object dynamics. In contrast, Qwen-VL-Max reaches only 15% in $S_{loc}$ but achieves 80% in grasp success rate ($S_{gra}$), suggesting some ability in finger closure but poor control of palm positioning. Seed-1.6 achieves the lowest $E_{loc}$ (0.66) and $E_{gra}$ (0.55). Although its overall localization success rate is not high, the low errors show stable geometric accuracy once localization is achieved. It also obtains the highest $S_{gra}$ (81%), reflecting strong local execution ability.

For trajectory quality, Qwen3-VL clearly outperforms the others. Its smoothness ($Q_{smooth} = 0.78$) and linearity ($Q_{line} = 0.85$) are close to the upper bound. This indicates stable trajectories with consistent direction and magnitude in continuous actions. Combined with its 30% localization success rate, the results suggest more robust performance in dynamic settings and highlight the advantage of large-scale VLMs in temporal consistency.

Finally, for completion speed ($R_{time}$), all models lag far behind GT, often completing localization only in later frames. This shows that current VLMs lack proactive prediction and early planning, and rely more on waiting for the target to enter a controllable region before executing grasping, which reduces overall efficiency.

## 5 CONCLUSION

We presented the first evaluation platform and benchmark dedicated to Dynamic Localization & Grasping: the **Dynamic Grasp Suite (DGS)** and the large-scale **Dyana-12M** benchmark. Our framework addresses the unique challenges of dynamic scenarios, moving targets, temporal alignment, and motion prediction, while supporting unified evaluation of VLAs, diffusion policies, and VLMs. Our results show that despite strong performance in static settings, current models still struggle to anticipate and act under dynamic conditions. By shifting evaluation from static to dynamic grasping, DGS and Dyana-12M provide valuable insights for guiding future research toward truly motion-aware hand intelligence.

**Ethics Statement.** We adhere to the ICLR Code of Ethics. Our work targets dynamic grasping in simulation and controlled robot testbeds; no human-subject studies were conducted, and no personally identifiable information (PII) is collected or released. The Dyana-12M benchmark is procedurally generated within our Dynamic Grasp Suite (DGS); assets used respect their original licenses, and no demographic attributes are included. Potential risks mainly concern unsafe deployment on real hardware; to mitigate this, we (i) evaluate primarily in simulation, (ii) provide safety notes for any optional hardware replication (emergency stop, workspace limits, collision checks), and (iii) release code and data under a research license with an explicit non-misuse clause. The benchmark does not aim to enable surveillance or other harmful applications. To our knowledge, there are no conflicts of interest or undisclosed sponsorship. The authors are responsible for all content and results reported.

**Reproducibility Statement.** We emphasize reproducibility through (i) an anonymized code release (supplementary materials) containing environment definitions, motion generators, evaluation scripts, and logging utilities; (ii) complete configuration files for each experiment (model/backbone, hyperparameters, random seeds, physics time-step, observation–action interface); (iii) deterministic data-generation scripts that recreate the Dyana-12M slices from a fixed seed and commit hash; (iv) formal metric definitions and implementation details in the appendix (success, endpoint errors, trajectory-quality scores, runtime); and (v) ablation protocols specifying training budgets and evaluation settings. Reference checkpoints and logs for baseline agents will be provided to verify tables in the main paper. Precise pointers to code folders, configs, and dataset preparation steps are included in the appendix to facilitate end-to-end replication.

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

# A    APPENDIX

## A.1    THE USE OF LARGE LANGUAGE MODELS (LLMS)

Our contribution is a comprehensive evaluation suite and benchmark for dynamic grasping. A primary evaluation target in our experiments is **vision–language models (VLMs)** and their **vision–language–action (VLA)** variants. We use DGS/Dyana to test and verify whether such models improve control under moving targets.

**No assistive LLMs for authorship or methodology.** We **did not use LLMs** for research ideation, paper writing/editing, algorithm design, code implementation, dataset creation/labeling, hyperparameter selection, or result interpretation. Any LLM/VLM/VLA appearing in the paper is a subject under evaluation, not a contributor to the research process.

**Originality, experiments, and accountability.** All methods (motion generators, metrics, and closed-loop protocol), software, and experiments were designed and executed by the authors without the use of LLM-generated content. We take full responsibility for all text, code, and results reported. This is consistent with the ICLR policy that LLMs are not eligible for authorship and undisclosed substantial LLM usage may be grounds for desk rejection.

**Reproducibility of evaluated LLM/VLM/VLA baselines.** When evaluating third-party VLM/VLA models, we document versions, checkpoints, prompts/interfaces (if any), and environment settings to ensure replicability. Our suite provides standardized observation–action APIs and seeds, allowing external researchers to reproduce the reported numbers.

**The appendix is organized as follows:**

- **Evaluation Protocol and Metrics** (Section A.2)

- **Motivation Claims** (Section A.3)

- **Trajectory Generation Details** (Section A.4)

- **Testing Details on Dynamic Grasp Suite** (Section A.5)

- **Theory Details** (Section A.6)

## A.2    EVALUATION PROTOCOL AND METRICS

When collecting dynamic grasping trajectories with DGS, we employ Unity's internal clock for environment stepping to ensure realistic and diverse sampling. During testing, the simulator is advanced manually at fixed intervals, and the target trajectory is kept identical across runs.

## A.3    MOTIVATION CLAIM

Current work on hand action generation mainly focuses on manipulating static objects, with limited models and benchmarks for dynamic object interaction. In static grasping tasks, evaluation can rely directly on ground-truth (GT) trajectories since the target remains fixed and a unique reference exists. In dynamic tasks, however, the agent may choose to grasp the moving object at different valid time points, and even if a GT is defined, it represents only one of many feasible solutions. This necessitates an interactive environment to assess whether a model can capture motion patterns and successfully grasp within the effective time window. To this end, we introduce the **Dynamic Grasp Suit (DGS)**, a unified platform for data generation and interactive evaluation in dynamic hand grasping.

## A.4 TRAJECTORY GENERATION DETAILS

DGS is built on the Unity engine and supports flexible, diverse dynamic data generation. We implement three fundamental motion scripts: linear, circular, and harmonic oscillation. Combined with a variety of collider components, these scripts can drive arbitrary 3D assets to move according to user-defined parameters, thereby producing highly diverse motion trajectories.

In addition, interception and localization-based grasping scripts enable a five-fingered hand to grasp arbitrary dynamic objects in real time. This process ensures both the continuity of object trajectories and the physical plausibility of hand motions, preventing unnatural or discontinuous action sequences. By composing motion scripts with grasping scripts, DGS can generate large-scale dynamic grasp trajectories that encompass diverse objects, motion patterns, and grasping styles, providing a rich data resource for training and evaluation.

## A.5 TESTING DETAILS ON DYNAMIC GRASP SUITE

During evaluation, the policy model communicates with the UHDS simulator in real time. At each time step, the simulator provides the latest observation, which is immediately forwarded to the model for inference, and the predicted action is then transmitted back to the simulator for execution.

Although we configure the simulator to maintain strict synchronization between the model and the simulated environment—ensuring that object and hand positions in the captured images are accurate—the real-time rendering process introduces subtle pixel-level variations across repeated trials at the same timestep. These variations are imperceptible to the human eye yet can slightly perturb the model's outputs. This setting imposes a stricter robustness requirement: while the critical information (object location and hand position) remains consistent, the model should not produce unstable or divergent actions due to visually negligible background differences.

**Evaluation Criteria**:

1. Localization success: The palm center is considered successfully localized if its projection on the $XZ$ plane lies within 0.3 units of the target's top surface. This threshold ensures that at least half of the object's area can be covered by the palm.

2. Grasping success: A grasp is regarded as successful if the predicted joint rotations reach at least 80% of the ground-truth (GT) values for all 15 joints, indicating that the hand outputs a grasping configuration sufficiently close to the GT action.

While Unity's real-time rendering engine ensures high physical realism, its inherent nondeterminism poses challenges to evaluation consistency. To address this, we employ Unity's internal clock for environment stepping during data collection, ensuring realistic yet diverse sampling. During model testing, the platform advances the simulator manually in a timed manner, synchronizing environment updates with model inference on the temporal axis. This mechanism guarantees time consistency between the environment and the model, enabling DGS to serve as a reliable online evaluation platform and providing a unified benchmark for dynamic hand grasping research.

## A.6 THEORY DETAILS

**Scope.** We study Hand-Object Interaction (HOI) with *moving* objects, a regime under-explored in prior HOI benchmarks that predominantly assume static or quasi-static targets. Our benchmark, **Dyana**, introduces motion-controlled targets whose trajectories are drawn from three primitive families: (1) straight-line (L), (2) circular-arc (A), and (3) simple-harmonic (H) motions. These primitives function as *motion tokens* that compose dynamic scenes with controllable difficulty.

**Notation.** A trajectory is a differentiable curve $\mathbf{r} : [0, T] \to \mathbb{R}^d$ with velocity $\mathbf{v}(t) = \dot{\mathbf{r}}(t)$ and acceleration $\mathbf{a}(t) = \ddot{\mathbf{r}}(t)$. For planar curves parameterized by time, curvature is

$$\kappa(t) = \frac{\|\mathbf{v}(t) \times \mathbf{a}(t)\|}{\|\mathbf{v}(t)\|^3}, \tag{1}$$

and in the arc-length parameter $s$ we use the Frenet-Serret frame $(\mathbf{T}, \mathbf{N}, \mathbf{B})$. For plane curves, the *fundamental theorem of curves* states that the curvature function $\kappa(s)$ determines the curve up to a rigid motion. We use $\tau(s)$ for torsion in 3D when needed.

**Signal-theoretic (harmonic) view.** Treat each coordinate of $\mathbf{r}(t)$ as a scalar time signal. If $x(t) \in L^2([0, T])$ is $T$-periodic, then by the Riesz-Fischer theorem its Fourier series converges in the $L^2$ sense:

$$x(t) = a_0 + \sum_{k=1}^{\infty} \big(a_k \cos(k\omega t) + b_k \sin(k\omega t)\big), \quad \omega = \tfrac{2\pi}{T}. \tag{2}$$

Hence $\mathbf{r}(t)$ can be approximated arbitrarily well (in energy) by a finite trigonometric polynomial. Uniform-frequency sine/cosine pairs with phase shift realize circular motions in $(x, y)$, the affine (zero-frequency) part gives linear drift. This motivates a small set of *frequency tokens* (dominant $\{k, A_k, \phi_k\}$ over local windows) to summarize oscillatory micro-motions under a controllable truncation budget.

**Geometric (curvature) view.** Because planar shape is encoded by $\kappa(s)$, approximating $\kappa(s)$ by a step function yields a piecewise constant-curvature curve, *i.e.*, a concatenation of straight segments ($\kappa \approx 0$) and circular arcs ($\kappa \approx 1/R$). This discretization aligns with nonholonomic optimal-control results: (i) **Dubins**: with bounded curvature and forward-only motion, the shortest path between two poses consists of at most three pieces from $\{L, R, S\}$ (left/right arcs and straight lines). (ii) **Reeds–Shepp**: allowing reversals, shortest paths admit finite canonical sequences of arcs/segments. When steering-rate continuity is required, *clothoids* (Euler spirals, linearly varying curvature) provide $G^2$-friendly connectors between $L/A$ pieces. These facts justify a compact *geometry-token* vocabulary with interpretable parameters (length, radius/sign, curvature slope). (Dubins, 1957; Reeds & Shepp, 1990; LaValle, 2006)

**Motion tokens and partial-order structure.** We represent trajectories as time-ordered tokens

$$\texttt{token} = (\text{type} \in \{L, A, C, \text{Harmonic}\}, \text{ params}, \Delta t),$$

where L=*line* ($\kappa \approx 0$), A=*arc* ($\kappa \approx \text{const}$), C=*clothoid* ($\kappa'$ nearly constant), and Harmonic encodes a dominant frequency component. Tokens live on intervals; across multiple DoFs we organize them with Allen's interval algebra (Before/Meets/Overlaps/During, *etc.*), yielding a DAG-like partial order that composes micro-motions into higher-level activities while preserving temporal constraints. (Allen, 1983)

## A.7 QUALITATIVE EXAMPLES OF SUCCESS AND FAILURE CASES

To complement the quantitative results in the main paper, we provide visual examples of successful and failed trajectories across three motion types: straight-line, circular-arc, and simple-harmonic. These examples illustrate the nuanced challenges posed by fast target motion, rapidly changing interception points, and the sensitivity of grasp timing. Each row in Figure A1 contains (left) a successful trajectory and (right) a representative failure case.

## A.8 FORMAL DEFINITION OF DYNAMIC LOCALIZATION & GRASPING

We formally define the task of *Dynamic Localization & Grasping* as a closed-loop hand–object interaction problem with a continuously moving target.

**Target motion.** Let the target object follow a continuous trajectory

$$r : [0, T] \to \mathbb{R}^3,$$

where $r(t)$ denotes the object center at time $t$, generated by a motion primitive from {linear, circular-arc, harmonic}.

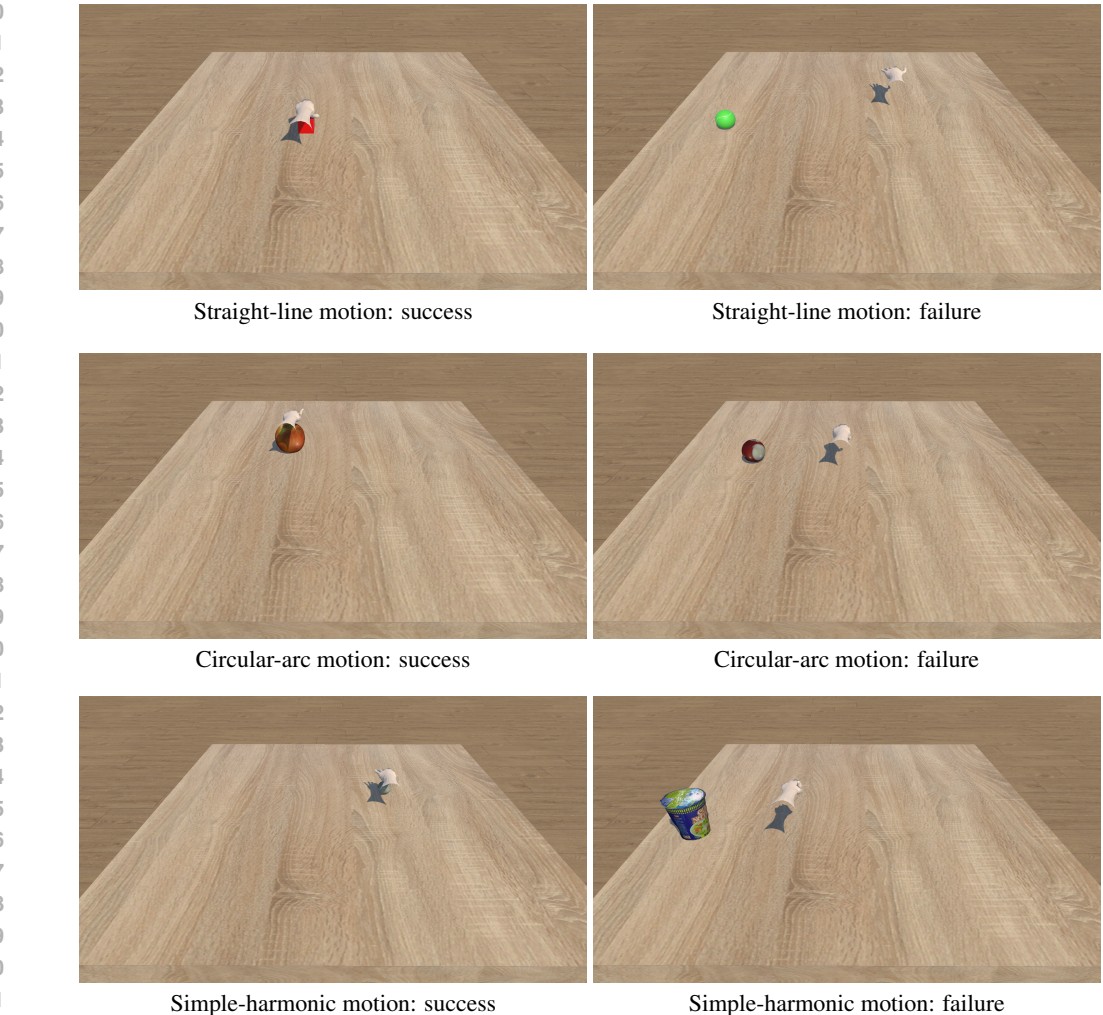

Straight-line motion: success          Straight-line motion: failure

Circular-arc motion: success           Circular-arc motion: failure

Simple-harmonic motion: success        Simple-harmonic motion: failure

Figure A1: Qualitative examples of successful (left) and failed (right) trajectories under three motion patterns: straight-line, circular-arc, and simple-harmonic.

**Observations.** At each time step $t$, the agent receives an observation

$$o_t = \{I_t,\ h_t,\ q_t,\ k_t,\ x^{\text{text}}\},$$

where $I_t$ is the RGB image, $h_t \in SE(3)$ is the palm pose, $q_t \in \mathbb{R}^{15}$ are the joint angles of the 15 finger DoFs, $k_t$ are fingertip keypoint poses, and $x^{\text{text}}$ is a task description.

**Actions.** A policy $\pi$ produces an 18-dimensional action

$$a_t = (a_t^{\text{loc}},\ a_t^{\text{gras}}),$$

where $a_t^{\text{loc}} \in \mathbb{R}^3$ updates the palm location and $a_t^{\text{gras}} \in \mathbb{R}^{15}$ updates the 15 joint angles. The executed hand trajectory is denoted by $\{p_t\}_{t=0}^T$, where $p_t$ is the palm center.

**Objective.** Dynamic Localization & Grasping is defined by the following conditions.

1. **Dynamic Localization.** The agent must choose a time $\tau \in [0, T]$ such that

$$\|p_\tau - r(\tau)\| \le \varepsilon_{\text{loc}},$$

where $\varepsilon_{\text{loc}}$ is a localization tolerance. The choice of $\tau$ is not predetermined: the agent must infer the motion of $r(t)$ and time its interception accordingly.

2. **Dynamic Grasping.** The agent must produce a joint configuration at (possibly different) time $\tau' \in [0, T]$ satisfying

$$\min_{f \in \mathcal{F}} \|f(q_{\tau'}) - r(\tau')\| \leq \varepsilon_{\text{gras}},$$

where $f(q_{\tau'})$ denotes fingertip surface positions induced by joint angles $q_{\tau'}$ and $\varepsilon_{\text{gras}}$ is a grasp tolerance.

3. **Closed-loop execution.** The agent must satisfy (1)–(2) while the target continues to move according to $r(t)$. Both $\tau$ and $\tau'$ depend on the inferred dynamics rather than being externally fixed.

**Success criteria.**   Dyana reports two *independent* task-level success measures: localization success and grasp success.

An episode achieves **localization success** if

$$\exists \tau \in [0, T] \quad \text{s.t.} \quad \|p_\tau - r_\tau\| \leq \varepsilon_{\text{loc}},$$

where $p_\tau$ denotes the palm-center position at time $\tau$.

An episode achieves **grasp success** if

$$\exists \tau' \in [0, T] \quad \text{s.t.} \quad \min_{f \in \mathcal{F}} \|f(q_{\tau'}) - r_{\tau'}\| \leq \varepsilon_{\text{gras}},$$

where $f(q_{\tau'})$ denotes the fingertip surface positions induced by joint configuration $q_{\tau'}$.

In addition to these two success indicators, the full palm trajectory $\{p_t\}_{t=0}^{T}$ is further evaluated using trajectory-quality metrics, including temporal smoothness, spatial linearity, and completion speed.

## A.9   Formal POMDP Formulation of Dynamic Localization & Grasping

We formalize Dynamic Localization & Grasping as a partially observable Markov decision process (POMDP)

$$\mathcal{M} = (\mathcal{S}, \ \mathcal{A}, \ \mathcal{O}, \ T, \ Z),$$

with the goal of learning a policy via supervised fine-tuning (SFT) on expert demonstrations rather than reinforcement learning.

**State space.**   A state $s_t \in \mathcal{S}$ represents the full physical configuration of the hand–object interaction at time $t$:

$$s_t = (r_t, \ v_t, \ h_t, \ q_t, \ k_t),$$

where $r_t \in \mathbb{R}^3$ is the object center, $v_t = \dot{r}_t$ its velocity, $h_t \in SE(3)$ the palm pose, $q_t \in \mathbb{R}^{15}$ the finger joint angles, and $k_t$ fingertip keypoint poses. The object evolves under a motion primitive (linear, circular-arc, or harmonic) with fixed parameters during an episode.

**Action space.**   The agent outputs an 18-dimensional control vector

$$a_t = (a_t^{\text{loc}}, \ a_t^{\text{gras}}) \in \mathcal{A},$$

where $a_t^{\text{loc}} \in \mathbb{R}^3$ updates the palm center and $a_t^{\text{gras}} \in \mathbb{R}^{15}$ updates the 15 controllable joints. These actions affect the next state through the simulator's physics engine.

**Transition model.**   The environment evolves according to

$$s_{t+1} = f_{\text{phys}}(s_t, a_t) + \xi_t,$$

where $f_{\text{phys}}$ denotes the Unity-based physics update and $\xi_t$ captures minor nondeterminism from real-time rendering. The object trajectory component $(r_{t+1}, v_{t+1})$ evolves independently of the agent's action according to the motion primitive governing the episode.

**Observation space.**   At each step, the agent receives a partially observed state

$$o_t = (I_t, \ h_t, \ q_t, \ k_t, \ x^{\text{text}}) \in \mathcal{O},$$

where $I_t$ is the RGB image and $(h_t, q_t, k_t)$ are proprioceptive features. The observation model $Z(o_t \mid s_t)$ is induced by the renderer and the simulator state.

**Policy.** The policy used in our benchmark is a vision–language–action (VLA) model, which maps the history of observations to an action:

$$a_t = \pi_\theta(o_{0:t}).$$

**Supervised learning objective (SFT).** Given an expert demonstration dataset

$$\mathcal{D} = \{(o_{0:T}^{(i)}, a_{0:T}^{(i)})\}_{i=1}^N,$$

collected using the Dynamic Grasp Suite, the policy parameters $\theta$ are optimized via supervised fine-tuning:

$$\theta^* = \arg\min_\theta \mathbb{E}_{(o_t, a_t) \sim \mathcal{D}} \big[ \mathcal{L}\big(\pi_\theta(o_{0:t}),\ a_t\big) \big],$$

where $\mathcal{L}$ is an action regression or distribution-matching loss depending on the VLA architecture.

**Task-level objective.** During evaluation, success is assessed by whether the executed trajectory satisfies both:

$$\|p_\tau - r_\tau\| \leq \varepsilon_{\text{loc}} \quad \text{(localization)}$$

for some $\tau \in [0, T]$, and

$$\min_{f \in \mathcal{F}} \|f(q_{\tau'}) - r_{\tau'}\| \leq \varepsilon_{\text{gras}} \quad \text{(grasping)}$$

for some $\tau' \in [0, T]$. Trajectory-level metrics (smoothness, linearity, completion timing) evaluate the entire executed trajectory $\{p_t\}$.

### A.10 SIMULATOR COMPARISON

To contextualize our choice of Unity as the simulation backend, we compare it with two widely used robotics simulators: **MuJoCo** and **IsaacSim**. The contrast is rooted in their fundamental design philosophies. MuJoCo and IsaacSim are engineered primarily for *robotics ecosystems*, offering high-fidelity physics, articulated robot models, and classical motion-planning tools. However, they provide almost no native support for **human hand skeletons**, **finger-level kinematics**, or **complex human motion**, which are essential for dynamic hand–object manipulation.

Furthermore, their core control stack is built around classical motion-planning algorithms (RRT, LQR, MPPI, CHOMP) combined with inverse kinematics solvers for robot arms. These planners assume *static or slowly moving* targets; extending them to support general-purpose grasping of *dynamically moving objects* requires substantial custom development and engineering overhead.

In contrast, Unity, as a general-purpose game engine, offers a rich asset ecosystem with **high-quality human hand models**, expressive animation pipelines, and convenient third-party motion libraries such as `DOTween` and `Spline`. More importantly, Unity exposes flexible and scriptable control interfaces, enabling a low-cost implementation of a full perception–action loop:

object motion $\rightarrow$ interception prediction $\rightarrow$ approach trajectory $\rightarrow$ arrival $\rightarrow$ grasping.

A single script can cover a wide range of dynamic object motions (linear, circular, projectile, irregular), making Unity particularly suitable for scalable data generation in our dynamic hand–object benchmark.

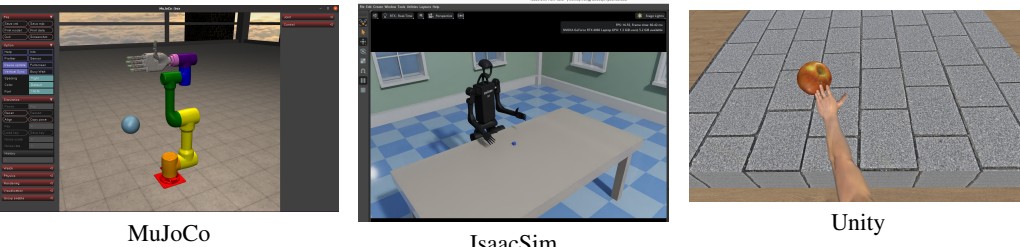

MuJoCo          IsaacSim          Unity

Figure A2: Visual comparison of MuJoCo, IsaacSim, and Unity environments used in our study.

## A.11 MORE COMPLEX MOTION

We evaluate the generalization of GR00T-N1.5 in three more challenging dynamic settings—projectile interception (Proj.), obstacle-aware planar grasping (ObsG.), and irregular-to-regular hybrid motion (Irreg.)—and observe that all models achieve near-zero success rates under these complex scenes.

Table A1: Performance of GR00T-N1.5 on three complex dynamic scenes—projectile motion (Proj.), obstacle-aware planar grasping (ObsG.), and irregular-to-regular motion (Irreg.).

| Dynamic Scene | $S_{\text{loc}}$ (%) ↑ | $S_{\text{gra}}$ (%) ↑ | $Q_{\text{smooth}}$ ↑ | $Q_{\text{line}}$ ↑ | $R_{\text{time}}$ ↑ |
|---|---|---|---|---|---|
| Proj. | 1.60 | — | 0.28 | 0.23 | 0.01 |
| ObsG. | **3.92** | **26.00** | 0.29 | **0.32** | 0.01 |
| Irreg. | 0.11 | — | **0.31** | 0.31 | 0.01 |

## A.12 ROBOT DOMAIN CASES

We additionally explored robotics-domain scenarios within Unity. The engine's flexible scripting interface allows direct joint-level control of a UR3 robotic arm model, enabling the execution of full grasping sequences such as reaching, pre-grasp alignment, and lifting. The control logic developed for dynamic hand motion generalizes naturally to robot manipulators, though additional engineering is required to adapt grasping strategies, trajectory generation, and timing policies to robotic embodiments.

Figure A3 presents two representative cases: **(i)** UR3 joint-space motion control; **(ii)** a gripper completing a grasp cycle from approach to lift. Each column shows a progression of states capturing the temporal evolution of the robot behavior.

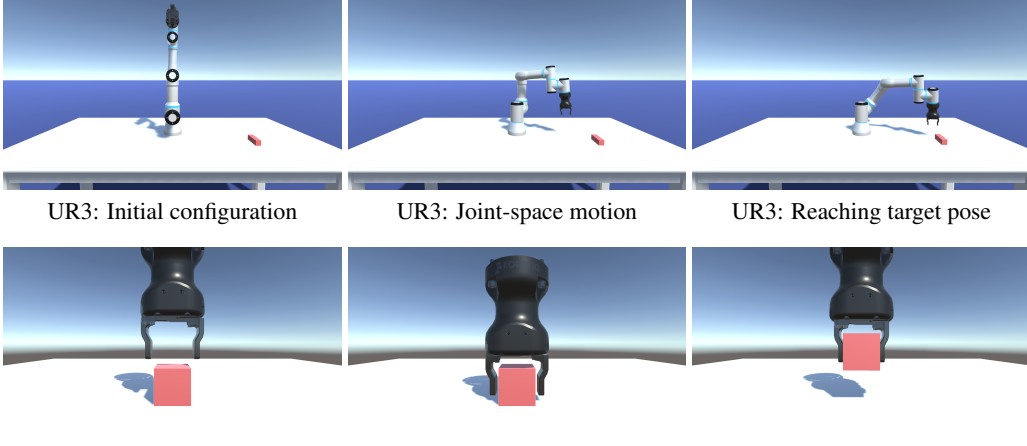

| UR3: Initial configuration | UR3: Joint-space motion | UR3: Reaching target pose |
|---|---|---|

| Gripper: Approach | Gripper: Closing and securing object | Gripper: Lift and hold |
|---|---|---|

Figure A3: Two robotic-domain examples implemented in Unity: (top row) UR3 arm joint-space control, and (bottom row) a gripper performing a full grasping sequence.

