# OpenReview forum: "Dyana: Benchmarking Dynamic Hand Intelligence"
_ICLR.cc/2026/Conference — Submitted to ICLR 2026_

### Official Review · Reviewer_6bHD · 2025-10-24

**Soundness:** 2
**Presentation:** 3
**Contribution:** 2
**Rating:** 4
**Confidence:** 3

**Summary:**

This paper introduces an evaluation framework for dynamic hand–object interaction and a large-scale dataset DYANA of human-hand grasp trajectories with targets following three interpretable motion primitives (line, arc, simple harmonic). The suite compares VLAs, diffusion policies, and VLMs under unified I/O and metrics.

**Strengths:**

- The paper is well organized and easy to understand.
- The motivation is good. The community lacks a unified benchmark for dynamic grasping tasks.
- The authors develop a large scale human-hand dynamic grasping dataset.
- The metrics are defined well

**Weaknesses:**

- I’m not fully convinced about benchmarking VLAs/diffusion policies together with VLMs that aren’t designed for robotics control. It risks mixing capability gaps with interface/latency quirks.
- The dataset is generated in Unity; visual robustness and sim2real transfer are critical issues.
- The diversity of the dataset is narrow. Motion comes from a small set of primitives and compositions; object/material variety is limited. It’s not obvious this covers the range of real-world grasp dynamics (curvature, accelerations, occlusions, camera motion, lighting, etc.).
- No sim2real experiments. There’s no real-robot study to show that rankings or trends in Dyana carry over. Without at least a correlation plot (sim metric vs. real success), it’s hard to trust transferability.
- Many VLAs/diffusion policies were trained on robot hands/grippers. Evaluating them on a human-like hand in Unity may measure adaptation pain more than “dynamic hand intelligence.”

**Questions:**

- Could you fix the caption issue in figure 1? There are two Grasp Deviation. I believe one of them should be Location Deviation.
- Do you have any real-robot results (even small-scale) showing that model rankings or metric deltas in Dyana predict real performance?
- Is DYANA an acronym? If so, please expand it on first use. If not, a one-line note on why you chose the name (and why the dataset is Dyana-12M) would help.

---

> ### Author Response · Authors · 2025-11-20
> **Response to reviewer 6bHD (part1)**
>
> Thank you for recognizing the motivation of our work and for providing detailed and constructive feedback. We greatly appreciate the opportunity to address your concerns and to further improve our manuscript.
>
> **W1: On jointly evaluating policy models and VLMs**
>
> > I’m not fully convinced about benchmarking VLAs/diffusion policies together with VLMs that aren’t designed for robotics control. It risks mixing capability gaps with interface/latency quirks.
>
> Our DGS platform enforces synchronous time progression between the model and the simulator. The simulator pauses until it receives the next model action, which removes interface and latency effects. The evaluation therefore isolates the model’s ability to infer actions from visual and linguistic observations.
>
>
>
> **W2: On visual robustness and sim realism**
>
> > The dataset is generated in Unity; visual robustness and sim2real transfer are critical issues.
>
> 1. This is indeed a challenge. Our goal is to address the lack of benchmarks and models for dynamic grasping of moving objects. Under DGS, we can at least evaluate the quality of trajectories generated online by policy models, rather than relying on framewise comparisons as in static tasks.
> 2. Our experiments show that current state-of-the-art models struggle even on simple motions. The benchmark therefore prioritizes high level dynamic-scene understanding; real-world deployment is outside the present scope.
>
>
>
> **W3: On data diversity**
>
> > The diversity of the dataset is narrow. Motion comes from a small set of primitives and compositions; object/material variety is limited. It’s not obvious this covers the range of real-world grasp dynamics (curvature, accelerations, occlusions, camera motion, lighting, etc.).
>
> 1. We explored diverse motion patterns, including projectile interception, obstacle-aware motion and transitions from irregular to regular motion. As shown in Table A1in Appendix A.11, existing models achieve near-zero success in these more difficult settings. Following the theory in the preliminaries (line 125) and Appendix A.6 (line 747), we focus on simpler atomic motions where current models still exhibit a performance gradient, which is essential for a meaningful benchmark. Additional visualizations about these complex motions are provided at the [link](https://drive.google.com/drive/folders/1wYBNDvGTRSJNjyugINmZJ3i2pfnWHbgk?usp=sharing).
> 2. The DGS platform supports rich dynamic-trajectory generation. Users can add new materials, backgrounds and motion parameters, and can also compose complex motion sequences. The unified grasping controller consistently produces physically valid hand trajectories for a wide range of moving objects.
>
>
>
> Table A1 in Appendix A.11: Performance	 of GR00T-N1.5 on three complex dynamic scenes—projectile motion (Proj.), obstacle-aware planar grasping (ObsG.), and irregular-to-regular motion (Irreg.).
>
> | Dynamic Scene | $S_{loc}$ (%) ↑ | $S_{gra}$ (%) ↑ | $Q_{smooth}$ ↑ | $Q_{line}$ ↑ | $R_{time}$ ↑ |
> | :-----------: | :-------------: | :-------------: | :------------: | :----------: | :----------: |
> |     Proj.     |      1.60       |        —        |      0.28      |     0.23     |     0.01     |
> |     ObsG.     |    **3.92**     |    **26.00**    |      0.29      |   **0.32**   |     0.01     |
> |    Irreg.     |      0.11       |        —        |    **0.31**    |     0.31     |     0.01     |
>
>
>
> **W4: On sim2real and the absence of physical robot tests**
>
> > No sim2real experiments. There’s no real-robot study to show that rankings or trends in Dyana carry over. Without at least a correlation plot (sim metric vs. real success), it’s hard to trust transferability.
>
> 1. **Although sim2real is important, it is not the focus of this work.** Our focus is on a more fundamental and forward-looking question: existing policy models exhibit limited capability in dynamic human–object interactions, and the field lacks a clear understanding of how well such models can reason about rapidly changing scenes. Our goal is therefore to study and improve their *high-level* understanding of dynamic environments, rather than to solve sim-to-real deployment. This benchmark serves as an early effort to clarify what current models can or cannot do in dynamic hand–object scenarios.
> 2. **We focus on hand motion generation rather than robotic manipulators**. The goal of our benchmark is to assess whether existing motion generation methods can handle moving targets and to evaluate their current capability, not to address the sim to real problem.

---

> > ### Author Response · Authors · 2025-11-20
> > **Response to reviewer 6bHD (part2)**
> >
> > **W5: On evaluating VLA and diffusion policies with a human hand**
> >
> > > Many VLAs/diffusion policies were trained on robot hands/grippers. Evaluating them on a human-like hand in Unity may measure adaptation pain more than “dynamic hand intelligence.”
> >
> > 1. VLA and diffusion policies are pretrained on large-scale datasets. Models such as GR00T-N1.5 and UP-VLA include substantial human-hand data and demonstrate strong performance across many static benchmarks.
> > 2. While task-specific architectures would perform better with sufficient domain training, dynamic human-hand grasping is under-explored and no specialized models exist for evaluation. We therefore use general-purpose VLA and diffusion policies, and we also fine-tune them on our data to assess their ability in hand-motion generation.
> >
> > **Q1: Correction in the teaser figure**
> >
> > > Could you fix the caption issue in figure 1? There are two Grasp Deviation. I believe one of them should be Location Deviation.
> >
> > We apologize for the oversight. The duplicated label Grasp Deviation has been corrected to Locate Deviation.
> >
> >
> >
> > **Q2: On robot testing and sim2real correlation**
> >
> > > Do you have any real-robot results (even small-scale) showing that model rankings or metric deltas in Dyana predict real performance?
> >
> > 1. This is an important point. Our DGS platform addresses the lack of benchmarks and models for dynamic grasping of moving objects. It enables fair and online evaluation of trajectory quality produced by policy models, rather than framewise alignment to ground truth as in static tasks.
> > 2. Our work should be viewed as a prototype and an initial exploration of a research space that is still largely undeveloped. The benchmark focuses specifically on human–object motion generation, rather than robotic manipulation, and is designed to establish a high-level understanding framework for dynamic scenes. Its purpose is to assess whether existing policy models can handle dynamic-object motion generation at all. The current state of the field is reflected in the weak performance of state-of-the-art models even on the simplest atomic motions, suggesting that drawing sim-to-real correlations at this stage would be premature. As model capability improves, systematic sim-to-real analysis will naturally become an important direction.
> >
> >
> >
> > **Q3: Meaning of DYANA**
> >
> > > Is DYANA an acronym? If so, please expand it on first use. If not, a one-line note on why you chose the name (and why the dataset is Dyana-12M) would help.
> >
> > The name is derived from Dynamic Hand Grasp Benchmark. Dyna inspires the reading Dyana.

---

### Official Review · Reviewer_K9ER · 2025-10-29

**Soundness:** 2
**Presentation:** 1
**Contribution:** 2
**Rating:** 2
**Confidence:** 4

**Summary:**

This paper addresses the problem of grasping moving objects, in particular the pre-grasp stage.
The main focus of this work is to studies how well current models can reach and grasp the moving objects.
To assess the grasping quality, this work proposes a few metrics, including completion time, pre-grasp trajectory smoothness and final grasp success rate.
The proposed benchmark generate simulated object motions, using a number of pre-defined basic motions.
To evaluate a model, e.g. VLA or Diffusion Model, the benchmark provides RGB observation to the model, and expect the model to output accurate global hand wrist pose and finger articulation.

**Strengths:**

This paper proposes to benchmark the grasping performance of moving objects. This problem setting is interesting and novel.

This paper studies a number of basic motions, these motions form a good basis for more complex motions.

This paper evaluates several state-of-the-art models on their benchmark, showcasing the limitation of current models.

**Weaknesses:**

# Reality Application
One issue of static object benchmark is whether such setting aligns well with real-life scenarios. For example, typically the objects are static w.r.t the environment before grasping. For this dynamic grasping benchmark, it is unclear how the benchmark performance will correlate to the real-life performance. The setting is novel but It is unclear in what specific robotics scenario we will need to grasp moving objects.

# Ambiguous Definition of Valid Grasps

The evaluation is centring around the successful  grasp, described in Line 348. However, the definition of “a valid grasp” is unclear.
Does a valid grasp account for objects sliding along fingers? Does it account for objects with non-uniform mass distribution? How are the forces evaluated?
Since the objects are in motion, what happens when a moving object hits the hand? Does the object bounce on the hand?

In real life, human will assess above questions before performing a successful grasp of a moving object. However, these factors are not considered in the proposed benchmark.

# Pre-grasp vs Grasping

The proposed evaluation metrics in Sec. 4.2 are primarily focusing on the pre-grasp stage: how fast the hand reaches the object, how smooth are the trajectory during reaching, whether the hand can locate above the objects. The only metric for grasping is the "Grasping success rate". It seems that this paper assume that the grasping is a solved problem once the hand is above the object, which I disagree.

In addition, "Grasping error" measures the distance between fingertips and the object surfaces at a successful grasp; this suggest that the validness of a grasp might merely be distance-based. However, a valid grasp is more than distance-based contact: "force closure" is another important factor affecting a grasp, and how forces interact with the object moving speed is not considered in the evaluation.

**Questions:**

I find this paper not easy to follow. Below are a few examples:

Line 49-53  the current evaluations are limited to static objects, but then the authors say “the blind spot … whether models can anticipate motion”. Does the Dyna benchmark evaluate methods on object motion forecasting?

Line 056, what does it mean to “expose” parameterized target-motion generators?

Line 056, what is observation-action API?

Line 066, what does rollout mean here?

Line 105: “CHOMP, ITOMP, TrajOpt” references needed.

Line 146: “Clothoids provide G2 connectors when needed”. What does this sentence mean?


In addition, I aslo found that this paper focuses on evaluating zero-shot methods, that is why VLA, DIffusion and VLMs are adopted for testing.
If the evaluation methods are not limited to zero-shot methods, we would expect a more traditional multi-stage method to work. By "traditional multi-stage" I mean training a depth estimator, a motion predictor and an explicit hand action model that output the global wrist and local finger poses.

---

> ### Author Response · Authors · 2025-11-20
> **Response to reviewer K9ER (part1)**
>
> Thank you for your thoughtful comments and for outlining the concerns you identified. We appreciate the opportunity to address these points and strengthen the clarity of our contribution.
>
> **W1: Reality Application**
>
> >One issue of static object benchmark is whether such setting aligns well with real-life scenarios. For example, typically the objects are static w.r.t the environment before grasping. For this dynamic grasping benchmark, it is unclear how the benchmark performance will correlate to the real-life performance. The setting is novel but It is unclear in what specific robotics scenario we will need to grasp moving objects.
>
> 1. **We first clarify that the domain we study is hand motion generation for dynamic objects.** In human-hand settings, predicting hand trajectories toward moving targets is important for VR and AR applications, where accurate prediction can support frame interpolation or path refinement to produce more realistic results. In robotic settings, industrial systems also must grasp objects moving on conveyor belts, and similar dynamic interactions arise in many service-robot scenarios.
> 2. **High-level dynamic-scene reasoning as the central challenge.** Current models perform poorly on dynamic grasping even under simple motions, suggesting that the primary challenge lies in high level understanding of dynamic scenes rather than in sim-to-real transfer. At this stage, advancing high level reasoning is more urgent than addressing cross-domain adaptation. Dyana-12M is therefore intended as a platform for evaluating dynamic grasping and for providing data that supports improving model capability in understanding and generating motion in dynamic environments.
>
>
>
> **W2: Ambiguous Definition of Valid Grasps**
>
> > The evaluation is centring around the successful grasp, described in Line 348. However, the definition of “a valid grasp” is unclear.
> >
> > 1. Does a valid grasp account for objects sliding along fingers?
> > 2. Does it account for objects with non-uniform mass distribution? How are the forces evaluated?
> > 3. Since the objects are in motion, what happens when a moving object hits the hand? Does the object bounce on the hand?
>
> 1. **Directly answer**：
>    1. The benchmark allows objects to slide across finger surfaces. Once contact occurs, the object’s motion is halted.
>    2. We do not simulate mass distribution or contact forces; grasps are evaluated based on geometric closure only.
>    3. Upon collision, moving objects do not bounce. The hand is treated as a non-rigid kinematic model, and object motion stops at contact.
> 2. **The design of the benchmark is shaped by the current performance limits of existing models.** Given that dynamic grasping is still at an early stage and policy models struggle significantly, we focus on motion-level understanding rather than fine-grained physical realism. Incorporating strict dynamics or detailed force evaluations would drive the performance of current models close to zero, so we emphasize motion diversity over precise force modeling.
>
>
>
> **W2_2: Assess above questions before perform**:
>
> > In real life, human will assess above questions before performing a successful grasp of a moving object. However, these factors are not considered in the proposed benchmark.
>
> We have considered this scenario. In fact, both the DGS platform and the Dyana-12M dataset support an observe-before-action mechanism, as described in Section 3.3 (line 206). At the beginning of each episode, the hand remains stationary while the object continues its predefined motion. This observation window allows the model to perceive temporal changes and learn motion patterns before executing any action.

---

> > ### Author Response · Authors · 2025-11-20
> > **Response to reviewer K9ER (part2)**
> >
> > **W3: Pre-grasp vs Grasping**
> >
> > > The proposed evaluation metrics in Sec. 4.2 are primarily focusing on the pre-grasp stage: how fast the hand reaches the object, how smooth are the trajectory during reaching, whether the hand can locate above the objects. The only metric for grasping is the "Grasping success rate". It seems that this paper assume that the grasping is a solved problem once the hand is above the object, which I disagree.
> >
> > 1. **Our goal is to evaluate dynamic scene understanding.** Given that current models already fail in simple settings, we simplify the grasping stage to ensure that the benchmark provides a measurable gradient instead of producing all zero results for more complex tasks. This creates a meaningful short term objective for the existing model zoo while leaving room for increased difficulty in future versions.
> > 2. **Evaluating human-like motion quality**：Because the embodiment is a human hand, we evaluate not only task completion but also the human likeness of the generated trajectories. This motivates our use of spatial linearity, temporal smoothness and completion speed as additional indicators.
> >
> >
> >
> > **W3_2: Grasping error**
> >
> > > In addition, "Grasping error" measures the distance between fingertips and the object surfaces at a successful grasp; this suggest that the validness of a grasp might merely be distance-based. However, a valid grasp is more than distance-based contact: "force closure" is another important factor affecting a grasp, and how forces interact with the object moving speed is not considered in the evaluation.
> >
> > The design of grasp error reflects limitations of current models rather than an ideal definition of grasp quality. Fully executing physically correct grasps on moving objects requires precise timing within a very narrow window, under which current policy models achieve near zero success. As a prototype benchmark, we provide a tractable intermediate objective instead of requiring full high fidelity grasp closure. This allows models to make measurable progress while the field develops stronger capabilities.

---

> > > ### Author Response · Authors · 2025-11-20
> > > **Response to reviewer K9ER (part3)**
> > >
> > > **Q1: Does Dyana evaluate explicit motion prediction?**
> > >
> > > > Line 49-53 the current evaluations are limited to static objects, but then the authors say “the blind spot … whether models can anticipate motion”. Does the Dyna benchmark evaluate methods on object motion forecasting?
> > >
> > > Dyana 12M does not evaluate motion prediction as an independent task. The term anticipate motion refers to whether a policy accounts for target dynamics while generating hand actions, for example by choosing appropriate interception points, rather than explicitly forecasting future trajectories.
> > >
> > >
> > >
> > > **Q2: Meaning of “expose the parameterized target motion generator”**
> > >
> > > > Line 056, what does it mean to “expose” parameterized target-motion generators?
> > >
> > > This phrasing was imprecise. It means that DGS provides a parameterized motion generator. Users can specify motion parameters such as radius, angular velocity or harmonic frequency, and the simulator produces a continuously moving target accordingly.
> > >
> > >
> > >
> > > **Q3: Meaning of the observation action API**
> > >
> > > > Line 056, what is observation-action API?
> > >
> > > It refers to the synchronized collection of image action pairs during rollout. The system aligns observations and actions in real time for the policy interface.
> > >
> > >
> > >
> > > **Q4: Meaning of rollout in line 66**
> > >
> > > > Line 066, what does rollout mean here?
> > >
> > > A rollout is the full execution of a policy in the environment from start to finish. We evaluate two settings: (1) the model outputs actions frame by frame based on multimodal inputs; (2) the model observes an initial segment without acting, then outputs continuous actions afterward. Dyana 12M includes an initial observation segment in every video, which users may keep or discard during training.
> > >
> > >
> > >
> > > **Q5: Missing references for CHOMP, ITOMP, TrajOpt**
> > >
> > > > Line 105: “CHOMP, ITOMP, TrajOpt” references needed.
> > >
> > > These references have been added in the revised manuscript.
> > >
> > >
> > >
> > > **Q6: Meaning of “Clothoids provide G2 connectors”**
> > >
> > > > Line 146: “Clothoids provide G2 connectors when needed”. What does this sentence mean?
> > >
> > > This means we use clothoid segments as smooth transition curves between motion primitives. Clothoids offer G2 continuity, ensuring continuity of position, tangent and curvature when linking straight segments, arcs and harmonic segments. Our benchmark uses these smooth connectors to maintain curvature continuity across motion primitives.
> > >
> > >
> > >
> > > **Q7: Why focus on zero shot VLA, diffusion and VLM models rather than multi stage pipelines**
> > >
> > > > In addition, I aslo found that this paper focuses on evaluating zero-shot methods, that is why VLA, DIffusion and VLMs are adopted for testing. If the evaluation methods are not limited to zero-shot methods, we would expect a more traditional multi-stage method to work. By "traditional multi-stage" I mean training a depth estimator, a motion predictor and an explicit hand action model that output the global wrist and local finger poses.
> > >
> > > We agree that a traditional multi stage pipeline combining depth estimation, motion prediction and explicit hand pose generation can in principle handle dynamic hand object interaction, and such task specific systems often perform strongly when tailored to a domain. However, dynamic localization and grasping is a new task and no dedicated multi stage systems currently exist. We therefore evaluate general purpose VLA, diffusion policies and VLM based controllers, and finetune them on Dyana 12M to align their action and observation spaces with our domain.

---

> > ### Comment · Reviewer_K9ER · 2025-11-25
> >
> > ### W1: Reality Application
> >
> > > In robotic settings, industrial systems also must grasp objects moving on conveyor belts
> >
> > I understand this, but it is still unclear how close Dyna motion resembles an actual conveyor belt motion. Currently Dyna motions are prototypes of those motions, and the authors directly claim that these prototypes are sufficient to represent real world dynamic motion. No further justification or evidence is given and thus I do not find this argument convincing.
> >
> >
> > ### Ambiguous Definition of Valid Grasps
> >
> > >  3. Upon collision, moving objects do not bounce. The hand is treated as a non-rigid kinematic model, and object motion stops at contact.
> >
> > I find the assumption “stopping object motion at contact” not fully convincing. If possible, having a few qualitative examples may better illustrate how this works; if this is a standard assumption, please show evidence from related works.
> >
> >
> > ### W2_2: Assess above questions before perform:
> >
> > Understood. Thanks for clarifying.
> >
> >
> > ————
> >
> > In my opinion, the concern about real-world applicability — the gap between Dyna motion and real-world motion — is the most crucial.

---

> > > ### Author Response · Authors · 2025-11-29
> > > **Response to reviewer K9ER (part1)**
> > >
> > > Thank you for your constructive feedback. We believe the following clarifications regarding the realism of our benchmark and the evaluation metrics will strengthen our work.
> > >
> > > ### **Response1_1: Realism of Hand Actions in Dyana**
> > >
> > > > I understand this, but it is still unclear how close Dyna motion resembles an actual conveyor belt motion.
> > >
> > > **Hand motions.**
> > >
> > > The actions involve hand translation and joint rotation. Please refer to the "Prototypes" folder in [link](https://www.google.com/url?sa=E&q=https%3A%2F%2Fdrive.google.com%2Fdrive%2Ffolders%2F1wYBNDvGTRSJNjyugINmZJ3i2pfnWHbgk%3Fusp%3Dsharing) for specific movement and grasping demos. The grasping action is a simple top-down grasp where the aperture correlates with the object scale, resembling how most humans pick up objects. Since the field of dynamic grasping is in its early stages, we did not design many variations of grasping actions. Instead, we focused on the diversity of target object motion to enhance the model's understanding of dynamic scenes.
> > >
> > > **Object dynamics.**
> > >
> > > The motion parameters are based on real-world scales. The 1m x 1m plane corresponds to the size of a standard manual workbench. Linear speeds range from 0.5 to 2.0 m/s, simulating industrial conveyor belts or daily scenarios of grasping moving objects. The circular motion radius of 10 to 36 cm reflects typical trajectories of small objects during manual interactions. The harmonic motion frequency of 1 to 4 Hz represents the common range for the swinging or shaking of lightweight objects in reality.
> > >
> > > **Hand structure.**
> > >
> > > We utilize Unity's kinematic model combined with realistic joint limits to ensure the structure closely resembles that of a human hand.
> > >
> > >
> > >
> > > ### **Response1_2: Why We Use the Three Atomic Motions**
> > >
> > > > Currently Dyna motions are prototypes of those motions, and the authors directly claim that these prototypes are sufficient to represent real world dynamic motion. No further justification or evidence is given and thus I do not find this argument convincing.
> > >
> > > 1. **Theoretical basis**: The rationale is detailed in Section 3.1 (L121) and Appendix §6 (L747).
> > >
> > > 2. **Trade-off between Benchmark Difficulty and Motion Diversity.** We explored extensive object motions as shown in the "Complex Motion" folder in [link](https://drive.google.com/drive/folders/1wYBNDvGTRSJNjyugINmZJ3i2pfnWHbgk?usp=sharing) and Table A1 in Appendix A.11. However, these proved too challenging since existing models only show performance gradients on simple planar motions. We avoided limiting our benchmark to simple planar motions by consulting theoretical foundations. **We utilized a minimal subset of motions including linear, curved, and harmonic movements to construct the benchmark. This design balances difficulty with motion diversity and realism.**
> > >
> > >    Table A1 in Appendix A.11: Performance of GR00T-N1.5 on three complex dynamic scenes: projectile motion (Proj.), obstacle-aware planar grasping (ObsG.), and irregular-to-regular motion (Irreg.).
> > >
> > >    | Dynamic Scene | $S_{loc}$ (%) ↑ | $S_{gra}$ (%) ↑ | $Q_{smooth}$ ↑ | $Q_{line}$ ↑ | $R_{time}$ ↑ |
> > >    | :-----------: | :-------------: | :-------------: | :------------: | :----------: | :----------: |
> > >    |     Proj.     |      1.60       |        —        |      0.28      |     0.23     |     0.01     |
> > >    |     ObsG.     |    **3.92**     |    **26.00**    |      0.29      |   **0.32**   |     0.01     |
> > >    |    Irreg.     |      0.11       |        —        |    **0.31**    |     0.31     |     0.01     |

---

> > > ### Author Response · Authors · 2025-11-29
> > > **Response to reviewer K9ER (part2)**
> > >
> > > ### **Response2: Ambiguous Definition of Valid Grasps**
> > >
> > > > I find the assumption “stopping object motion at contact” not fully convincing. If possible, having a few qualitative examples may better illustrate how this works; if this is a standard assumption, please show evidence from related works.
> > >
> > > We clarify the meaning of *“stopping object motion at contact”* and our motivation for adopting this setting, based strictly on the ground-truth (GT) data collection process.
> > >
> > > 1. **What “stopping object motion at contact” means**: During data collection, the GT grasping procedure follows a standard human-like dynamic interception routine: observe the moving object, predict its intercept point, move toward that point, wait until the object enters the palm region, **press the palm downward to stop the object’s motion**, and then close the hand. This full process is shown in the *Prototypes* folder of the [project videos](https://drive.google.com/drive/folders/1wYBNDvGTRSJNjyugINmZJ3i2pfnWHbgk?usp=sharing)
> > >
> > > 2. **Why we adopt this setting in evaluation**: Most existing VLA models cannot even reliably execute the simplest part of this GT behavior, **namely lowering the palm to “pin” a moving object when it is already under the hand.** Because this limitation makes full dynamic grasp execution currently infeasible, we decompose the evaluation into localization accuracy and grasp accuracy, and accordingly reduce the overall difficulty.
> > >
> > >    1. **Localization accuracy**: Localization accuracy does **not** require the model to perform the palm-down stopping action. Instead, we only check whether the object is already under the predicted palm position. In real applications, this moment could be detected by rule-based heuristics and followed by an external controller that drives the palm-down action if needed.
> > >
> > >    2. **Grasp accuracy (semantic correctness of the grasp)**: Grasp accuracy evaluates whether the model outputs a **semantically valid grasping motion**, independent of object stopping dynamics. As stated in Appendix A.5 (line 735), a grasp is considered successful if the joint rotations output by the model approximate the GT configuration closely
> > >
> > >       > Excerpt (line 735):
> > >       >
> > >       > Grasping success: A grasp is regarded as successful if the predicted joint rotations reach at least 80% of the ground-truth (GT) values for all 15 joints, indicating that the hand outputs a grasping configuration sufficiently close to the GT action.
> > >
> > >    3. **Alignment with the goal of our work**: This design reflects the core purpose of our benchmark, which is to **evaluating whether models can generate human-intuitive hand actions in dynamic scenes,** not sim-to-real performance. Because dynamic grasping of moving objects remains extremely challenging for today’s model zoo, our primary focus is on action generation quality and semantic interpretability, rather than requiring full physical execution of the GT stopping behavior.

---

### Official Review · Reviewer_YqX9 · 2025-10-29

**Soundness:** 2
**Presentation:** 1
**Contribution:** 2
**Rating:** 2
**Confidence:** 4

**Summary:**

This paper aims to address the limitations of existing benchmarks that focus on static objects and introduce a new framework for benchmarking dynamic hand intelligence. The authors propose DGS, a simulation platform built in Unity, and Dyana-12M, a large-scale dataset of 12 million frames featuring human-hand trajectories for grasping objects moving along predefined atomic paths. The framework includes standardized evaluation protocols and hierarchical metrics to assess performance. The authors evaluate several state-of-the-art policy models and VLMs to show that these models struggle with dynamic grasping tasks.

**Strengths:**

- The proposed Dyana-12M dataset is extensive and incorporates features tailored specifically for dynamic scenarios. The "Observe-before-act" mechanism is a novel and critical feature for evaluating a model's ability to infer motion patterns before execution.
- The hierarchical evaluation metrics, which cover success rates, trajectory quality, and completion speed, are comprehensive.
- The paper provides a valuable empirical study by benchmarking a wide range of recent and relevant models.

**Weaknesses:**

- The manuscript suffers from numerous contradictions, typos, and undefined terms that severely impede readability and trust in the results. The camera setup is described as providing "egocentric RGB images... from a fixed camera" (line 158). These terms are mutually exclusive. An egocentric camera moves with the agent, while a fixed camera does not. Figure 1 contains two separate axes with different values labeled "Grasp Deviation," making the chart ambiguous and difficult to interpret. Key acronyms are used without definition. For instance, "HOI" is used multiple times before it is ever defined, forcing the reader to guess its meaning (Hand-Object Interaction).
- The paper fails to provide a formal definition of the task and does not adequately justify key modeling assumptions. The task of "Dynamic Localization & Grasping" is never formally defined. The paper lacks a clear mathematical formulation (e.g., as a POMDP) specifying the state, action, observation spaces, and the objective function. This makes it difficult to understand the precise problem the models are supposed to solve. Moreover, the benchmark assumes a "free-floating (hand-centric) embodiment". This abstracts away the robot arm, which is a major source of kinematic and dynamic constraints in any real-world application. This assumption is not discussed or justified, and it raises concerns about the practical relevance of the benchmark, as some evaluated trajectories (e.g., Fig. 2) might be physically unrealizable for a robot.
- Key design choices for the simulation environment and trajectory representation are not well-defended. The paper states that DGS is implemented in the Unity engine, but provides no justification for this choice over other simulators like Isaac Sim or MuJoCo, which are more common in robotics research and often offer better physics fidelity and sim-to-real support. The motivation for using linear, arc, and harmonic motion primitives is brief. The claim that these "compose into arbitrarily complex trajectories" is very strong and not sufficiently supported. Details about "For models from different ... through lightweight adaptation" (lines 201-204) are also not discussed.
- The paper relies exclusively on quantitative tables and charts, providing no visual examples of the generated trajectories. For a paper introducing a new benchmark and simulation environment, showing visual examples of success and failure cases is critical to demonstrate the nuances and challenges of the task.

**Questions:**

See weaknesses.

---

> ### Author Response · Authors · 2025-11-20
> **Response to reviewer YqX9 (part1)**
>
> Thank you for the detailed feedback. We understand the concerns you raised, and we are grateful for the opportunity to clarify these aspects and provide further evidence supporting our design choices.
>
> **W1_1: Wording issues and figure errors**
>
> > The manuscript suffers from numerous contradictions, typos, and undefined terms that severely impede readability and trust in the results. The camera setup is described as providing "egocentric RGB images... from a fixed camera" (line 158). These terms are mutually exclusive. An egocentric camera moves with the agent, while a fixed camera does not. Figure 1 contains two separate axes with different values labeled "Grasp Deviation," making the chart ambiguous and difficult to interpret. Key acronyms are used without definition. For instance, "HOI" is used multiple times before it is ever defined, forcing the reader to guess its meaning (Hand-Object Interaction).
>
> We apologize for the imprecise wording and duplicated labels.
>
> 1. The camera is egocentric, meaning it maintains alignment with the hand object interaction. To keep the hand consistently in view, the camera automatically moves backward when the hand is too close and forward when it is farther away.
> 2. In Figure 1, the duplicated Grasp Deviation has been corrected to Location Deviation.
> 3. The term HOI is now clearly defined as Human Object Interaction. All corrections have been updated in the revised PDF.
>
> **W2_1: Formal definition of the task**
>
> > The paper fails to provide a formal definition of the task and does not adequately justify key modeling assumptions. The task of "Dynamic Localization & Grasping" is never formally defined. The paper lacks a clear mathematical formulation (e.g., as a POMDP) specifying the state, action, observation spaces, and the objective function. This makes it difficult to understand the precise problem the models are supposed to solve.
>
> 1. We provide a formal definition of Dynamic Localization and Grasping in Appendix A.8. Localization and grasping are treated as two stages because direct grasp evaluation leads to extremely low success rates under current model capabilities. For grasping, we therefore evaluate whether the model produces effective joint rotations.
> 2. Appendix A.9 also includes a MDP formulation that defines the state, action, observation and objective in a precise manner.
>
>
>
> **W2_2: On the free floating hand centric embodiment**
>
> > Moreover, the benchmark assumes a "free-floating (hand-centric) embodiment". This abstracts away the robot arm, which is a major source of kinematic and dynamic constraints in any real-world application. This assumption is not discussed or justified, and it raises concerns about the practical relevance of the benchmark, as some evaluated trajectories (e.g., Fig. 2) might be physically unrealizable for a robot.
>
> 1. **Our work is a prototype aimed at an emerging task.** Because current state of the art models already struggle on simple dynamic motions, we focus on improving high level understanding rather than low level deployment constraints.
> 2. Therefore, as an early contribution in this area, **we intentionally focus on the human hand**, whose motion strategy is naturally more flexible than that of robotic manipulators. This choice maximizes the freedom of dynamic interactions and allows the benchmark to capture the **general principles of human hand grasping under dynamic scenes**. use a hand centric embodiment that maximizes kinematic freedom and exposes the core challenge of dynamic hand object interaction.
> 3. As shown in Appendix A12, **we also include a robotic-arm demonstration to illustrate that our framework can, in principle, extend to different embodiments**. However, robotic control introduces additional engineering considerations and is not the focus of this work. The benchmark is designed specifically to evaluate hand motion generation under dynamic targets, rather than robotic manipulation or sim-to-real transfer. At the current stage , we center our evaluation on the human-hand domain, which best reflects the core objective of DGS.

---

> > ### Author Response · Authors · 2025-11-20
> > **Response to reviewer YqX9 (part2)**
> >
> > **W3: On the choice of Unity instead of Isaac Sim or MuJoCo**
> >
> > > Key design choices for the simulation environment and trajectory representation are not well-defended. The paper states that DGS is implemented in the Unity engine, but provides no justification for this choice over other simulators like Isaac Sim or MuJoCo, which are more common in robotics research and often offer better physics fidelity and sim-to-real support.
> >
> > In Appendix A.10, we provide additional illustrations that explain in detail why we choose Unity. The reasons can be summarized into two main points.
> >
> > 1. **Physics engines**. MuJoCo is a model based engine tailored for articulated robots and end effectors. It lacks animation, camera, material and skeleton systems, making it unsuitable for human hand domains. Unity and Isaac Sim both use NVIDIA PhysX, a highly general engine. Unity provides mature skeleton animation and built in hand rigs, which are essential for dexterous hand grasping.
> > 2. **Dynamic scene support**. MuJoCo and Isaac Sim center on motion planning and inverse kinematics for static targets. To support moving object grasping, one must implement a multi degree hand model, dynamic interception prediction, a hand specific motion planner and grasp closure logic. These components are non trivial. Unity, in contrast, offers rich hand models, an expressive animation pipeline, convenient motion control libraries (e.g. DOTween, Spline) and a scriptable interface that allows a single script to cover object motion prediction, interception, approach and grasping.
> >
> > **W3_2: On the use of three atomic motion primitives**
> >
> > > The motivation for using linear, arc, and harmonic motion primitives is brief. The claim that these "compose into arbitrarily complex trajectories" is very strong and not sufficiently supported.
> >
> > **We evaluated more complex cases indeed**. Visualizations are provided in the [link](https://drive.google.com/drive/folders/1wYBNDvGTRSJNjyugINmZJ3i2pfnWHbgk?usp=sharing) and quantitative results in Table A1 in Appendix A.11. Existing models achieve near zero performance in these settings. To provide a meaningful benchmark and offer current models a clear target to strive for, we first focus on simple atomic motions. The preliminaries (line 125) and Appendix A.6 (line 747) in the paper also explain how complex motions can be decomposed into these primitives. As models improve, the benchmark will include more difficult tasks.
> >
> > Table A1 in Appendix A.11: Performance of GR00T-N1.5 on three complex dynamic scenes: projectile motion (Proj.), obstacle-aware planar grasping (ObsG.), and irregular-to-regular motion (Irreg.).
> >
> > | Dynamic Scene | $S_{loc}$ (%) ↑ | $S_{gra}$ (%) ↑ | $Q_{smooth}$ ↑ | $Q_{line}$ ↑ | $R_{time}$ ↑ |
> > | :-----------: | :-------------: | :-------------: | :------------: | :----------: | :----------: |
> > |     Proj.     |      1.60       |        —        |      0.28      |     0.23     |     0.01     |
> > |     ObsG.     |    **3.92**     |    **26.00**    |      0.29      |   **0.32**   |     0.01     |
> > |    Irreg.     |      0.11       |        —        |    **0.31**    |     0.31     |     0.01     |
> >
> >
> >
> >
> >
> > **W3_3: Clarification on lightweight adaptation**
> >
> > > Details about "For models from different ... through lightweight adaptation" (lines 201-204) are also not discussed.
> >
> > The adaptation refers to finetuning the evaluated models on the Dyana 12M training set to align their action space and observation space with the human hand dynamic grasping domain. We revised the wording for clarity.
> >
> >
> >
> > **W4: Visualization results**
> >
> > > The paper relies exclusively on quantitative tables and charts, providing no visual examples of the generated trajectories. For a paper introducing a new benchmark and simulation environment, showing visual examples of success and failure cases is critical to demonstrate the nuances and challenges of the task.
> >
> > Appendix A.7 now includes success and failure cases for all three atomic motions to provide direct visual understanding of model behavior.

---

### Official Review · Reviewer_ghd6 · 2025-11-01

**Soundness:** 2
**Presentation:** 2
**Contribution:** 3
**Rating:** 6
**Confidence:** 1

**Summary:**

This paper proposes a benchmark for hand grasping for dynamic objects, which can capture dynamic, real-world scenarios where the grasp targets are dynamic -- in contrast to prior hand grasping benchmarks focused on static objects. To this end, it introduces the Dynamic Grasp suite (DGS), which is a unified, online evaluation platform for dynamic grasp with parameterized motion generations. It also presents Dyna-12M benchmark containing 12M frames across 180K dynamic hand grasp trajectories, constructed on top of DGS. The paper also provide standardized evaluation for diverse model faimilies including VLA agents, diffusion policies and VLMs.

**Strengths:**

**(1) Contributing a novel and useful benchmark**

This paper introduces a large-scale benchmark and evaluation platform for hand grasping of dynamic objects — an underexplored area in the field. I believe this can be a valuable contribution to the research community. I particularly appreciate the authors’ effort to release an online evaluation platform and provide standardized evaluations for existing model suites, which will further enhance the accessibility and usability of this benchmark.

**(2) Reproducibility**

The authors provide an implementation of the proposed benchmark, which significantly improves reproducibility.

**(3) Good presentation quality**

Overall, the presentation quality of this paper is solid. The figures are well-designed, and the text is clearly organized and easy to follow.

**Weaknesses:**

**(1) Real–synthetic domain gap of motions**

The proposed framework employs parameterized target-motion generation, enabling automated data collection and annotation, as demonstrated in the dataset comparisons in Table 2. However, this motion parameterization inherently limits the expressiveness of the generated motions, which may limit its ability to model complex real-world dynamics. Additional explanations or discussions on this aspect would be helpful.

**(2) Lack of discussion on limitations**

Related to the above point, the paper lacks sufficient discussion of the current limitations of the proposed benchmark. Including such a discussion would provide more informative insights to the research community and better guide future research directions.

**Questions:**

Please see the weaknesses section. I especially wonder how significant the real–synthetic domain gap is in the generated motions.

---

> ### Author Response · Authors · 2025-11-20
> **Response to reviewer ghd6**
>
> We sincerely appreciate your recognition of the novelty of our work and your detailed, constructive comments. Your feedback is highly valuable, and we welcome the opportunity to clarify and strengthen our submission.
>
> **W1: On the real–synthetic motion gap**
>
> > The proposed framework employs parameterized target-motion generation, enabling automated data collection and annotation, as demonstrated in the dataset comparisons in Table 2. However, this motion parameterization inherently limits the expressiveness of the generated motions, which may limit its ability to model complex real-world dynamics. Additional explanations or discussions on this aspect would be helpful.
>
> 1.  **On using parameterized motions**. Parameterization does simplify real-world dynamics, but it makes evaluation tractable and avoids the difficulty of collecting repeatable trajectories for moving objects. Real-world data acquisition for dynamic grasping remains highly challenging.
> 2. **On the effect of this simplification.** Our experiments show that state of the art models already fail on these simplified motion patterns. This indicates that the core difficulty lies not in the physical complexity of the motion but in the dynamic-scene reasoning required to succeed.
> 3.  **On future directions.** We view this benchmark as a necessary foundation. Once models demonstrate consistent competence on these controlled settings, extending the task to richer and more complex real-world motions will be a natural and important next stage.
>
>
>
> **W2: On the limitations**
>
> > Related to the above point, the paper lacks sufficient discussion of the current limitations of the proposed benchmark. Including such a discussion would provide more informative insights to the research community and better guide future research directions.
>
> 1. **Limited variation in appearance and materials.** Our benchmark prioritizes motion diversity and uses large-scale parametric variation to generate rich dynamic trajectories. Compared with this broad motion coverage, the current version contains a narrower range of object appearance and material properties. Although multiple shapes and sizes are included, complex materials such as transparent, deformable or highly reflective objects are not yet systematically represented.
> 2.  **Single-modality perception.** The dataset uses single-view RGB observations with fixed camera poses. This design isolates dynamic trajectory understanding but simplifies perception challenges relative to real environments, where multi-camera fusion, viewpoint drift and hand–object occlusion are more common.
> 3. **Limited human-behavior variability.** The evaluation metrics introduce temporal smoothness, spatial linearity and completion speed to encourage human-like motion. However, the automated control scripts used to guarantee physical feasibility produce trajectories that are consistently smooth and linear. Human behaviors such as hesitation, compensatory micro-adjustments or re-planning after failed attempts are not yet modeled.
>
> **Q1: sim2real gap**
>
> > Please see the weaknesses section. I especially wonder how significant the real–synthetic domain gap is in the generated motions.
>
> The synthetic motion patterns in Dyana focus on simple and well structured trajectories. For these motions, the gap between real and simulated domains is small because many real world dynamic interactions are dominated by such first order patterns, for example linear motion on conveyor belts or near circular arcs in everyday handoffs. More complex real world trajectories may introduce larger gaps, but they are often composed of segments that resemble these simple primitives. Once models can reliably handle these primitives, extending the benchmark to richer composite motions will be the next step.

---

### Meta-Review · Area_Chair_uT5u · 2026-01-02

**Summary:**

The paper initially receives scores of 6, 2, 2, and 4. The AC has carefully read the entire paper, along with all reviews and rebuttals, and acknowledges the major concerns raised by the reviewers, particularly that the manuscript is not yet ready, the task definition and evaluation are ambiguous, and the sim-to-real gap remains insufficiently addressed.

Therefore, the AC recommends rejection. The AC also suggests that the authors carefully revise the paper by addressing the reviewers’ comments.

**Reviewer Concerns:**

The reviewers raised several concerns, with the key and common issues summarized below.

* Reviewer YqX9 and K9ER note that the manuscript contains numerous contradictions, typos, and undefined terms, making it difficult to follow. They also point out the lack of visual examples of generated trajectories. The rebuttal corrects some issues and provides partial clarification.
* Reviewer YqX9 and K9ER argue that the paper fails to provide a formal definition of the task and its assumptions, with ambiguous definitions of valid grasps and unclear evaluation metrics for pre-grasping. The rebuttal refers to definitions provided in the Appendix and clarifies that the work focuses on motion-level understanding rather than fine-grained physical realism.
* Reviewer ghd6, 6bHD, and K9ER raise concerns about the synthetic-to-real domain gap, sim-to-real transfer, and the lack of real-world applications or sim-to-real experiments. The rebuttal argues that even the simplified motion patterns in the synthetic setting are already highly challenging, and that many real-world dynamic interactions are dominated by similar first-order motion patterns, suggesting a limited domain gap.
* Reviewer XqX9 questions whether key design choices in the simulation environment and trajectory representation are sufficiently justified. The rebuttal explains that these choices are motivated by the availability of mature skeleton animation systems and support for dynamic scenes.
* Reviewer 6bHD points out that the diversity of the dataset is limited and questions whether it adequately covers real-world grasp dynamics. The rebuttal states that different motion patterns are included and notes that existing models achieve near-zero success in more difficult settings.

For the first two concerns, the AC believes that the manuscript requires substantial revision. Since the core contribution of the paper is the definition of a new task and its evaluation protocol, clear and precise definitions are critical for reproducibility and for enabling meaningful follow-up work.

Regarding the gap between synthetic dynamic motion and real-world motion (Concern 3), the AC finds that the rebuttal does not fully address or convincingly resolve the reviewers’ concerns. In particular, the rationale for using simplified synthetic motion and the extent of its gap from real-world motion need to be explicitly discussed and supported by empirical evidence.

**Reviewer Scores:**

The AC expects that most reviewers will likely maintain their initial scores, as the concerns raised by Reviewers YqX9, K9ER, and 6bHD regarding manuscript quality, task definition, and the sim-to-real gap require substantial revision and are difficult to be fully addressed during the rebuttal stage. In addition, Reviewer ghd6 reports a low confidence score of 1.

---

### Decision · Program_Chairs · 2026-01-26

Reject